# A network analysis to identify mediators of germline-driven differences in breast cancer prognosis

Maria Escala-Garcia et al.[#]

Identifying the underlying genetic drivers of the heritability of breast cancer prognosis remains elusive. We adapt a network-based approach to handle underpowered complex datasets to provide new insights into the potential function of germline variants in breast cancer prognosis. This network-based analysis studies ~7.3 million variants in 84,457 breast cancer patients in relation to breast cancer survival and confirms the results on 12,381 independent patients. Aggregating the prognostic effects of genetic variants across multiple genes, we identify four gene modules associated with survival in estrogen receptor (ER)-negative and one in ER-positive disease. The modules show biological enrichment for cancer-related processes such as G-alpha signaling, circadian clock, angiogenesis, and Rho-GTPases in apoptosis.

[#]A full list of authors and their affiliations appears at the end of the paper.

Family-based studies have suggested that breast cancer survival in first-degree relatives has a hereditary component[1,2]. Nevertheless, whereas large-scale genome-wide association studies (GWASs) have made considerable progress in identifying germline variants linked to breast cancer risk[3], the identification of germline variants linked to breast cancer prognosis has proven more challenging[4]. An understanding of how and which germline variants affect breast cancer prognosis could provide novel insights into the etiology of the metastatic process in breast cancer, increase knowledge on the underlying heterogeneity of the disease, and help identify new therapeutic targets or select patients most likely to benefit from existing therapies.

A major limitation of the studies to date is that the sample sizes have been insufficient to detect the small effect sizes of germline variants characteristic for breast cancer risk and survival[4–6]. Even though our previous survival GWAS included >95,000 patients[4,5], the limiting factor was the relatively low number of events (breast cancer-specific deaths) observed. One way to overcome this limited power is to use pathway or network-based approaches[7,8]. These techniques typically use predefined gene sets, annotated pathways, or protein–protein interaction (PPI) networks to detect genetic effects across multiple genes or proteins with similar or related biological functions[6,8–10]. Using such methods, a biological pathway might emerge as relevant even if none of its individual germline variants reached genome-wide significance. Moreover, assigning the variants to genes reduces dimensionality: considering several pathways as opposed to millions of individual variants leads to a substantial reduction in the number of tests performed[11]. An additional advantage of performing a pathway analysis is that it naturally suggests which biological processes mediate the genetic association with survival, making the biological interpretation easier[7,11–13].

Here we report on a network-based GWAS to identify genetic determinants of breast cancer prognosis in a dataset with a total of 84,457 breast cancer patients of European ancestry. In line with previous studies, we did not find many individual genetic variants with strong effects[14–17]. However, aggregating the survival estimates of multiple variants across genes and using a network propagation method, we identified several biological processes that may mediate a germline genetic effect on breast cancer prognosis. These include key processes in cancer biology, such as regulation of apoptosis, G-alpha signaling, and the circadian clock mechanism. In our analysis, we show that the identified polygenic effects are associated with survival not only in the discovery set but also in an independent dataset of 12,381 patients. In addition, we studied the downstream transcriptional changes and their functional consequences due to the prognostic variants. We observed similar biological processes in the enrichment of the downstream and module-level gene analyses suggesting that both levels are perturbed by the identified genetic variants.

## Results

### Single variant and gene analyses detect one independent hit.
We performed an analysis of the association between germline genetic variants and breast cancer prognosis comprising data for 84,457 female breast cancer patients of European ancestry. To account for potential subtype-specific associations, we also performed separate analyses for estrogen receptor (ER)-positive and ER-negative breast cancer. An overview of all data is given in the "Methods" section and Supplementary Table 1. As a first step in our analysis, we tested the association of ~7.3 million imputed genetic variants with breast cancer-specific survival using a Cox proportional hazard model (Fig. 1a). Based on a genome-wide statistical significance $P$ value threshold of $5 \times 10^{-8}$, we identified

two variants at 8q13, in high linkage disequilibrium (LD) with each other, associated with survival in ER-positive breast cancer. The top variant was rs6990375 (chr8:70571531, $P = 6.35 \times 10^{-9}$) followed by rs13272847 (chr8:70573316, $P = 1.07 \times 10^{-8}$). We did not find significant variants for ER-negative or all breast cancer cases.

Next, we aggregated the summary statistics of the individual variants into gene-level $P$ values (~21,800 genes in total) using the Pascal algorithm[12] (Fig. 1b). We computed the gene score based on the maximum chi-squared signal within a window size of 50-kb around the gene region (see "Methods"; Fig. 2). Two genes were associated with survival in ER-positive breast cancer at $P < 0.05$ after Bonferroni correction: *SLCO5A1* ($P = 4 \times 10^{-7}$, corrected $P = 0.01$) and *SULF1* ($P = 7 \times 10^{-7}$, corrected $P = 0.02$) (Fig. 2c). These two genes are located in close proximity to each other around the significant variants at 8q13 identified in the single variant analysis. Their significance is therefore likely driven by a single causal genetic variant. The top variant rs6990375 is situated in the 3′ untranslated region of *SULF1* where it may affect the binding of regulatory micro-RNAs. While the association of this variant with breast cancer survival has not been identified previously, it has been reported to be associated with age of onset of ovarian cancer[18]. *SULF1* has been found to be involved in cell proliferation, migration, and invasion as well as drug-induced apoptosis in cancer cell lines[19], most likely due to its regulatory role in fibroblast growth factor[20] and Wnt signaling[21]. Less is known about the function of *SLCO5A1*, although a role in cell proliferation has been suggested[22]. In line with the single variant analysis, we found no significant genes for all breast cancer or ER-negative breast cancer (Fig. 2a, b) when aggregating individual variants into genes.

### Network analysis finds germline-related prognostic modules (GRPMs).
To explore whether weaker signals of association were hidden in our data, we investigated the hypothesis that the germline genetic variants associated with breast cancer prognosis target particular biological processes but within those processes do not uniquely target one particular gene. Different subgroups of patients might harbor variants in different genes, which ultimately affect the same biological process. Such polygenic signals, unless they have very big effects, may remain undetected if only individual variants or even individual genes are tested. We therefore applied network propagation[23], a technique that maps gene association scores onto a PPI network and uses the network topology to detect sub-networks, or modules, of closely interacting, high-scoring proteins (Fig. 1c). In the context of this paper, we will refer to these modules also as germline-related prognostic modules (GRPMs).

For the network propagation, we used the HotNet2 method[13], which has been used previously with GWAS data[24]. We based the gene scores on the aggregate gene $P$ values computed by the Pascal method (see "Methods"). The protein interaction network used by HotNet2 was obtained from iRefIndex[25].

When considering all breast cancers, the HotNet2 analysis identified no significant GRPMs (lowest $P = 0.06$, based on the HotNet2 permutation test). In contrast, several GRPMs were associated with prognosis in the analyses by ER subtype. For ER-positive patients, the best HotNet2 result ($P$ value <0.01) comprised 31 GRPMs of ≥7 genes. For ER-negative patients, the best HotNet2 results ($P < 0.01$) included 116 GRPMs of ≥4 genes. A list of all significant prognostic modules is presented in Supplementary Data 1.

To help the interpretation of the identified GRPMs, we developed an extension to HotNet2 that maps the module genes

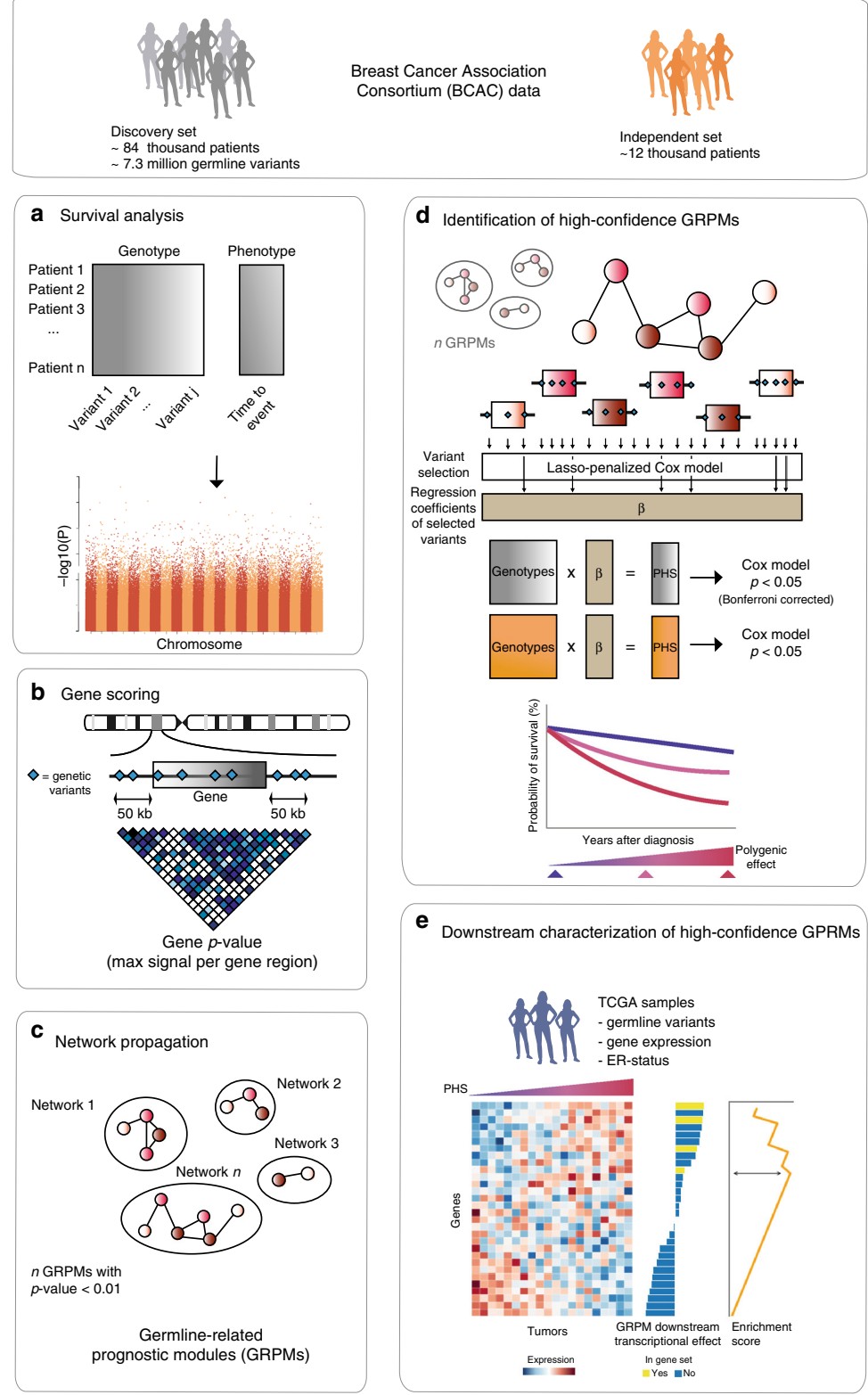

to the specific genetic variants that are most strongly associated with prognosis. This was done by performing a Lasso-penalized Cox regression on the genetic variants assigned to the module genes. Using those selected variants and their effect sizes, a polygenic hazard score (PHS) was computed and used to identify a set of high-confidence GRPMs (Fig. 1d), as well as to perform a functional characterization of the downstream effects of the prognostic variants (Fig. 1e).

**Prognostic modules point to underlying pathways.** We restricted our scope to a subset of high-confidence GRPMs. This subset was identified by testing the association of each module's PHS with breast cancer prognosis in an independent set of 12,381 patients (with 1120 events) (Supplementary Table 2) that was not used previously in the HotNet2 analysis or in the construction of the PHS score. GRPMs with a significant association between PHS and prognosis (P value <0.05, based on a one-sided Wald

**Fig. 1 Network analysis pipeline (see "Methods" for details). a** Cox models were used to estimate the association between each genetic variant and breast cancer-specific survival in 84,457 patients of the Breast Cancer Association Consortium (BCAC) dataset (discovery set). **b** The P values of the survival analyses for the genetic variants (blue diamonds) were used to compute gene scores using the Pascal algorithm. These gene scores were based on the maximum chi-squared signal within a window size of 50-kb around the gene region and accounted for linkage disequilibrium structure (depicted in a gradient blue scale). **c** The HotNet2 method was used to identify gene modules based on the $-\log_{10} P$ value of the computed gene scores. **d** The modules found by HotNet2 were filtered to obtain a selection of high-confidence germline-related prognostic modules (GRPMs). We constructed a polygenic hazard score (PHS) summarizing the prognostic effects of a set of selected genetic variants in the module. We then tested the association of this PHS with survival in both the discovery set (gray) and the independent set (orange). **e** We performed a functional characterization of the high-confidence GRPMs by studying the downstream transcriptional effects. For that, we used genotype and expression data from The Cancer Genome Atlas (TCGA). We computed the correlation between a GRPM's polygenic hazard score and the expression of all available genes. Based on these correlation values, a gene set enrichment analysis assigned biological processes that were enriched among the genes most correlated with the prognostic variants in the GRPM.

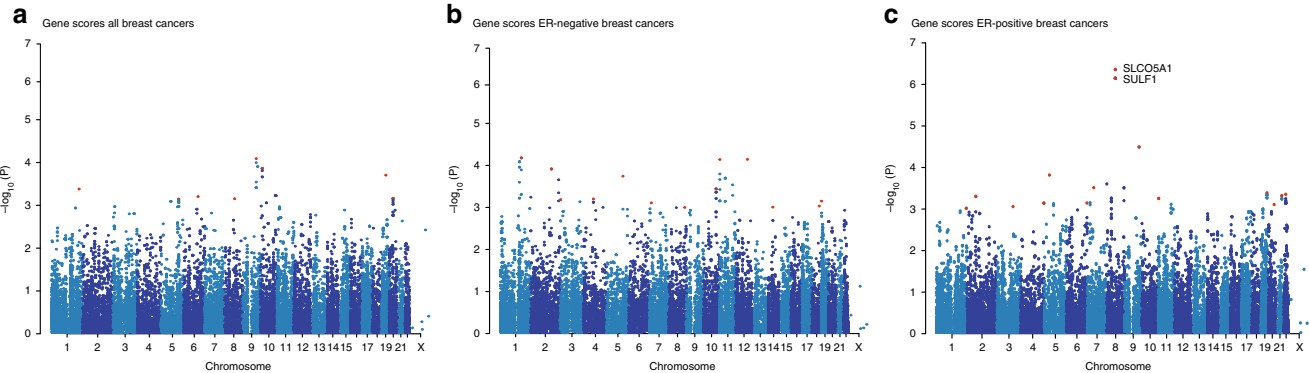

**Fig. 2 Manhattan plots of the gene-level associations with breast cancer-specific survival.** Plots show the association in **a** all breast cancer cases ($n =$ 84,457), **b** estrogen receptor (ER)-negative ($n = 14,529$), and **c** ER-positive ($n = 55,701$). The $-\log_{10}$ gene P values from the Pascal algorithm is shown on the y axis and genomic position on the x axis. The top significant genes and the most significant gene per chromosome (if $-\log_{10}(P) > 3$) are shown in red.

test) in this independent set were considered high confidence. Following this procedure, we found four high-confidence GRPMs for ER-negative breast cancer (Fig. 3a–c) and one high-confidence GRPM for ER-positive breast cancer (Fig. 3d). Hazard ratios of the association of the PHSs with breast cancer-specific survival ranged from 1.09 to 1.28 (Fig. 3e). In the remainder of this section, we will discuss the high-confidence GRPMs. The term PHS P value will be used to refer to the P value of a GRPM's PHS association with survival.

To provide a functional characterization of the five high-confidence GRPMs found in the ER-negative and ER-positive subtypes, we tested each module for enriched biological processes on two levels. The first, which we call the module-level, considers the direct functions of the GRPM proteins themselves. These were identified by an enrichment analysis of the annotated biological functions of the module proteins and their direct interactors in a PPI network annotation (see "Methods"). For the high-confidence GRPMs in ER-negative breast cancer, we identified enriched processes related to G-alpha signaling, cell growth, and angiogenesis; insulin secretion; and circadian clock (Supplementary Fig. 1a–d). For the ER-positive high-confidence GRPM, the enriched processes included signaling by Rho GTPases and apoptosis (Supplementary Fig. 1e).

The module-level enrichment provides a general summary of the biological functions of the GRPM genes. However, it is based on functional annotations that have been derived from studies in many different cell types and biological environments. To study the specific downstream effects of the identified prognostic variants in breast cancer tumors, we performed enrichment analyses on the downstream transcriptional changes due to the prognostic variants affecting the module proteins.

We estimated these downstream transcriptional effects using genetic variants and RNA expression data of female breast cancer patients from The Cancer Genome Atlas (TCGA)[26]. For each of the five GRPMs, the downstream analysis was performed on the subset of TCGA patients matching the ER subtype in which the GRPM was identified, 118 patients with ER-negative and 440 with ER-positive tumors. Using the germline genotype data of these TCGA patients, we computed the PHS for each GRPM (Supplementary Table 3). Based on these PHSs, we then computed GRPM downstream transcriptional effect scores, which reflect the correlation between a module's PHS and the mRNA expression level of every gene (Fig. 1e; see "Methods"). Using the obtained downstream transcriptional effect scores, we performed gene set enrichment analysis (GSEA)[27] with gene sets based on Reactome[28] and the MSigDB[29] Hallmark gene sets. The enrichment results for the MSigDB Hallmark gene sets are shown in Fig. 3, only pathways with a GSEA P value <0.001 and false discovery rate (FDR) < 0.01 were included in the visualization. The full list of enriched processes per high-confidence GRPM can be found in Supplementary Data 2–6 and Supplementary Fig. 2.

The enriched pathways in the downstream analysis included biological processes, such as cell cycle, DNA repair, metabolism of RNA, lipids or proteins, apoptosis, and translation of proteins. Importantly, we observed overlap of the biological processes enriched in the downstream analysis and those found for the module proteins. This observation has two important implications. First, it provides additional support for the biological role assigned to the module proteins. In addition to this, in cases where module proteins may serve several roles, it helps identify which of those roles is affected by the prognostic variants at a transcriptional level. The enriched biological processes assigned to the modules and the related downstream processes are described below.

*ER-negative: G-alpha signaling events*: Two high-confidence GRPMs found for patients with ER-negative tumors (Fig. 3a) suggested, from the module-level analysis, G-alpha signaling and

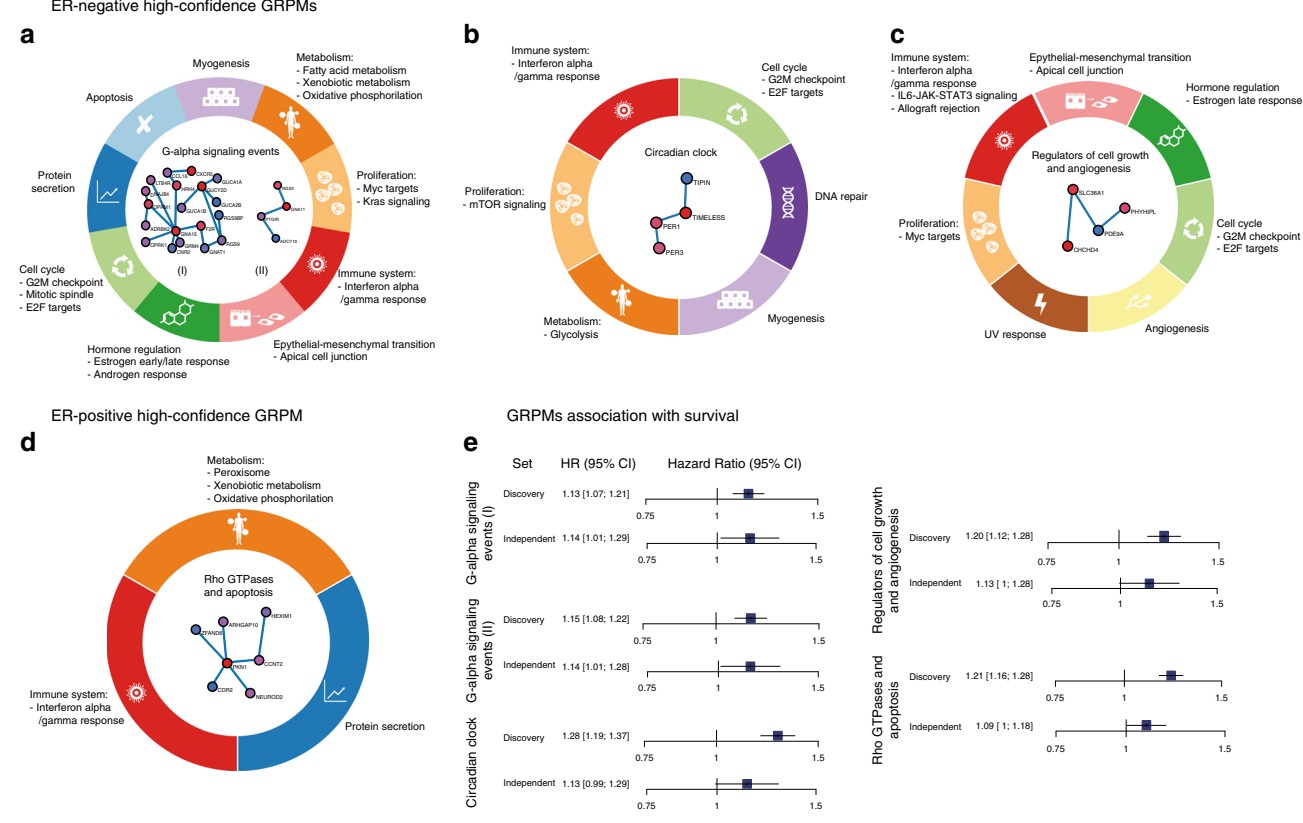

**Fig. 3 High-confidence germline-related prognostic modules (GRPMs).** The GRPM is shown at the center of the circles, surrounded by the biological processes enriched among the downstream transcriptional effects of each module. Three modules were found for estrogen receptor (ER)-negative breast cancer (**a**–**c**) and one module was found for ER-positive breast cancer (**d**). **a** G-alpha signaling GRPMs. **b** Circadian clock GRPM. **c** Regulators of cell growth and angiogenesis GRPM. **d** Rho GTPases and apoptosis GRPM. **e** Plots illustrating the association between each GRPM's PHS and 10-year breast cancer specific-survival in the discovery and independent sets. HR hazard ratio (per standard deviation of the PHS), CI confidence interval. The error bars show the 95% confidence interval. The confidence intervals shown are two sided, whereas the significance test performed was one sided (see "Methods").

G-protein activation as biological processes associated with survival. The first GRPM (PHS $P = 0.0096$) includes *ADCY10*, *GNA11*, *PTGIR*, and *RGS3* (Fig. 3a, right) and the other GRPM (PHS $P = 0.0082$) is a larger module of 19 genes: *ADRBK2*, *CCL16*, *CNR2*, *CXCR5*, *DNAJB4*, *F2R*, *GNA15*, *GNAT1*, *GRM4*, *GUCA1A*, *GUCA1B*, *GUCA2B*, *GUCY2D*, *HRH4*, *LTB4R*, *OPRK1*, *OPRM1*, *RGS9*, and *RGS9BP* (Fig. 3a, left).

On closer inspection of the genetic variants selected for the two modules' PHSs, we observed that one genetic variant was shared by both modules. The other variants in the PHSs, 2 variants in total for the 4-gene module and 3 variants for the module of 19 genes, were also located in the same genomic region on chromosome 19p13.3 (Fig. 4a). These variants are upstream of *GNA11* in the former module and *GNA15* in the latter. For the other genes in these two GRPMs, no genetic variants were selected as part of the modules' PHSs. This may be due to lack of statistical power: although the gene scores were high enough to be included in the module, none of their individual genetic variants had a strong enough association. The co-location of *GNA11* and *GNA15* provides an explanation for why the identified variants were selected for both modules. It also suggests that the genetic associations of these two genes and hence of the two modules are not independent. Indeed, the patients' PHSs for both GRPMs are highly correlated (Fig. 4b), which supports a shared genetic association. This raises the question of whether the putative germline genetic effect on survival is mediated through both genes or only one of the two. In the downstream analyses of both modules, changes of *GNA15* expression were identified as one of the strongest downstream transcriptional effects, whereas this is

not the case for *GNA11*. Conversely, in an independent gene expression dataset using KMplotter (http://kmplot.com/analysis), we found that expression of *GNA11* is significantly associated with recurrence-free survival in ER-negative breast cancer (Supplementary Fig. 3), while a similar effect was not seen for *GNA15*. These preliminary observations leave open the hypothesis of a role for both genes. A definitive answer will require more functional analyses.

In the module-level analysis, the GRPM formed by four genes also showed enrichment for insulin secretion. It has been shown that there is a close relationship between G-proteins and their coupled receptors (GPCR), insulin, and the insulin-like growth factor I receptor. Altered versions of this crosstalk could play a role in cancer cells[30,31]. For example, it has been proposed that, in cancer cells, insulin can increase the activity of GPCRs in cancer tissues via the mTOR (mammalian target of rapamycin) pathway[31], which was also one of the enriched processes in the downstream analysis. The highest scoring gene in the module, *GNA11*, codes for the alpha subunit of the $G_{11}$ protein, which has been linked to insulin secretion and signaling[32,33].

For the 19-gene GRPM, we also identified thrombin signaling and platelet aggregation as two of the main module-level enriched pathways. Thrombin is a type of the above-mentioned GPCRs with the capacity to upregulate genes able to induce, or contribute to oncogenesis and angiogenesis, and is known to be able to stimulate the adhesion of tumor cells to platelets[34]. In the downstream analysis, we identified processes such as GPCR ligand binding and hemostasis, which contributes to the thrombosis process and therefore is also linked to GPCRs[35]

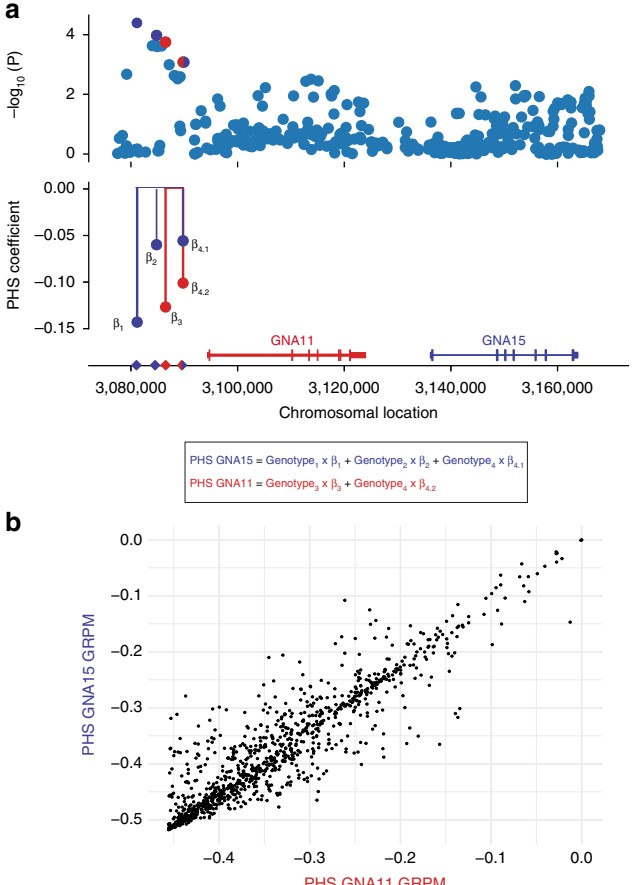

**Fig. 4 Genomic region 19p13.3 with the two genes *GNA11* and *GNA15*.** The two G-alpha signaling high-confidence germline-related prognostic modules (GRPMs) identified in the estrogen receptor (ER)-negative subtype have a shared genetic signal in the same genomic region. **a** Top: $-\log_{10}(P)$ for the association with survival (y axis) of all variants in the region 19p13.3 (y axis). Bottom: regression coefficients from the survival model for the genetic variants in the module's polygenic hazard scores (PHSs). **b** Scatter plot comparing the two modules' PHSs in the iCOGS independent validation set. PHS of the *GNA11* GRPM on the x axis and PHS of the *GNA15* GRPM on the y axis.

(Supplementary Fig. 2a and Supplementary Data 2). It has been reported that hemostatic elements such as platelets, coagulation, and the fibrinolytic system might play an important role in breast cancer progression and metastasis[36].

*ER-negative: circadian clock*: Another module identified by our network analysis consists of four genes with a strong link to the circadian clock mechanism: *PER1*, *PER3*, *TIMELESS*, and *TIPIN* (PHS $P = 0.030$; Fig. 3b). Having an important role in the regulation of the cell cycle[37], the circadian clock is believed to be important in the development of cancer. Disrupted sleep patterns and associated changes to the body's circadian rhythm have long been implicated in the risk of developing several cancers, including breast cancer[37–39]. Although long-term night-shift work has not consistently been found to be associated with breast cancer[40], one study reported an increased risk of ER-negative breast cancer[41]. More recently, genetic variants in circadian clock genes have been reported to be associated with breast cancer risk[42,43]. In addition to risk, the circadian clock has also been suggested to be involved in breast cancer progression and prognosis[44,45].

More specifically, the circadian clock genes in this module have also individually been implicated in the biology of cancer in general and breast cancer in particular. The period genes *PER1* and *PER3* have been found to suppress cancer cell growth[46,47] and have also been observed to be deregulated in breast cancer[48]. *TIMELESS* and its interactor *TIPIN* are believed to be central players in the connection between the circadian clock and the cell cycle and apoptosis[49,50]. The importance of these genes in the regulation of the cell cycle was supported by the downstream analysis, which pointed out that cell cycle-related processes are strongly enriched among the downstream transcriptional changes.

*ER-negative: regulators of cell growth and angiogenesis*: The last high-confidence GRPM identified for ER-negative breast cancer contains proteins that have been linked to regulation of cell growth or angiogenesis: *CHCHD4*, *PDE9A*, *SLC36A1*, and *PHYHIPL* (PHS $P = 0.027$; Fig. 3c). Knock down of *CHCHD4* has been found to reduce tumor growth and angiogenesis in vivo[51]. In addition, *CHCHD4* has been observed to mediate the mitochondrial translocation of p53[52] through which it may trigger apoptosis via the p53 mitochondrial pathway[53]. *PDE9A* is a regulator of cGMP signaling, a pathway that is increasingly being recognized as an important player in breast cancer biology[54]. Inhibition of *PDE9A* has been found to trigger apoptosis in both ER-positive and ER-negative breast cancer cell lines[55]. *SLC36A1*, also known as *PAT1*, has been linked to tumor cell growth through its involvement in the activation of mTORC1. *PHYHIPL* (or *PAHX-AP1*) has mostly been described in the context of neuronal cells, but no role in cancer has been described.

*ER-positive: Rho GTPases in apoptosis and cell growth*: For ER-positive tumors, we identified one high-confidence module (PHS $P = 0.020$; Fig. 3d). The module was predicted to be involved in Rho GTPases effectors, which typically function as binary switches controlling a variety of biological processes. Because of their ability to control cell motility, they have been hypothesized to play a role in progression and metastatic dissemination of cancer cells[56]. This GRPM contains seven genes: *ARHGAP10*, *CCNT2*, *CDR2*, *HEXIM1*, *NEUROD2*, *PKN1*, and *ZFAND6*. *ARHGAP10* (rho GTPase Activating Protein 10) was previously reported as the most significant locus ($P = 2.3 \times 10^{-7}$) in a GWAS of breast cancer survival[14]. The top scoring gene in the module, *PKN1* (protein-kinase-C-related kinase), controls processes such as regulation of the intermediate filaments of the actin cytoskeleton, tumor cell invasion, and cell migration[57]. It is activated by the Rho family of small G-proteins and might mediate the Rho-dependent signaling pathway[58], which was one of the main enriched pathways in the module-level analysis. *PKN1* has also been described as an important player in other cancers: in androgen-associated prostate cancer by controlling migration and metastasis[57] or in melanomas by inhibiting Wnt/β-catenin signaling and apoptosis[58].

From the module-level analysis, another enriched main process was the pathway linked to *PTEN* (phosphatase and tensin homolog deleted on chromosome 10) regulation, which is a well-characterized tumor suppressor[59]. *PTEN* is directly involved in the metabolism of phospholipids and lipoproteins[60], adaptive immune system, and B cell receptor associated events,[61] which were all hits in the downstream analysis. One of the six genes in the module, *HEXIM1* (hexamethylene bisacetamide-inducible protein 1), is a positive regulator of p53 and has been identified as a potential novel therapeutic target modulating cell death in breast cancer cells[62]. In the downstream analysis of this module, we also identified processes present in the module-level analysis that highlighted key tumorigenic biological processes (Supplementary Data 6), for instance, pathways related to p53 activity, Wnt signaling, regulation of mRNA stability by proteins that bind AU-rich elements, or apoptotic execution phase.

## Discussion

There is evidence that breast cancer prognosis has a heritable component[2,63,64]. Exploring the possible link between germline genetic variants and breast cancer survival may help to develop better criteria for breast cancer stratification, which might have implications for breast cancer prognostication and treatment[65]. However, identifying germline genetic variants associated with breast cancer prognosis has been challenging so far, mainly because the current sample sizes have been insufficient to detect small effect signals.

In this work, we started with a survival analysis based on individual germline variants similar to the previous GWAS we have undertaken[4]. While in the previous analyses no variants reached genome-wide significance, here we identified two genome-wide significant variants for ER-positive tumors (rs6990375: $P < 6.35 \times 10^{-9}$ and rs13272847: $P = 1.07 \times 10^{-8}$) located in 8q13. More complete follow-up and more conservative variant filtering per dataset (only including variants with imputation $r^2 > 0.8$) may have enabled identification of these variants that remained below genome-wide significance in our previous study ($P = 3.02 \times 10^{-5}$ and $P = 1.73 \times 10^{-5}$, respectively). In the gene-level analysis, we found two significant genes (*SLCO5A1* and *SULF1*, $P < 0.05$ after Bonferroni correction) associated with breast cancer survival. It is likely that both associations were driven by the identified leading variant rs6990375.

To address the lack of power in the individual germline variant and gene-level analyses, we developed a network analysis method that revealed five high-confidence GRPMs associated with breast cancer prognosis. We identified four modules specific for ER-negative breast cancer and one for ER-positive breast cancer. The GRPMs comprise crucial processes such as cell cycle and progression, regulation of apoptosis, signaling by mTOR, immune system, G-alpha signaling, and the circadian clock. These processes are already known to play a role in cancer biology in general and breast cancer prognosis specifically. However, our results highlight the possible regulatory impact of germline variants on these processes, which traditionally has received little attention in cancer survival studies. The broad range of genes and functions seems to indicate, as already hypothesized, that breast cancer survival is a complex phenotype influenced by many factors and biological mechanisms.

The analysis by ER status subtypes identified significant associations that were not present when analyzing all patients together. This is in line with the breast cancer risk analyses undertaken in this same dataset, where the ER subtype analyses also identified new associations[3]. In addition, the main classification of breast cancer tumors used for prognosis and treatment selection is based on immunohistochemical markers such as ER, progesterone receptor, and HER2 status, reflecting the fact that each group has a different etiology and prognosis. This assumption is further supported by a comparison of the gene association scores between the ER status subtypes. The gene scores for ER-positive and ER-negative breast cancer are uncorrelated (Supplementary Fig. 4c) (Pearson correlation = −0.002), while the gene scores for all breast cancer cases seem to resemble the ER-positive subtype more (Supplementary Fig. 4a; Pearson correlation = 0.366) than the ER-negative subtype (Supplementary Fig. 4b; Pearson correlation = 0.197). In addition, we found that the distribution of PHSs across patients was similar for ER-positive and ER-negative breast cancer patients (Supplementary Fig. 5), but importantly, each PHS was associated with prognosis only for the subtype in which it was found (Supplementary Table 4). These differential associations across subtypes suggest that prognosis is inherited differently for these two different disease classes.

The network-based approach and the stratification of patients by ER status enabled a refined interpretation of the GWAS results[5,66], but the findings are still limited due to the number of deaths observed, limited follow-up, missing treatment information, and possibly remaining heterogeneity of tumor subtype within the ER classes. Increased sensitivity and specificity of the results could be achieved by including additional patients and by adjusting for more fine-grained tumor characteristics and the treatment received. Moreover, the network propagation results are dependent on the completeness of the PPI network used. As a notable consequence of this, we did not identify modules containing the two gene-level significant hits *SLCO5A1* and *SULF1*, due to the fact that the PPI network did not contain the proteins they code for.

The modules that are identified also depend on the specificity of the PPI network to the disease-relevant tissue. Many proteins have tissue-specific expression patterns and functions; hence not all interactions in a generic PPI network are found in all tissues. The use of a tissue-specific PPI network may prevent discovery of false positive modules. One single most relevant tissue for our analysis is not easily identified though. Unlike the somatic mutations found in tumor cells, the germline variants we studied are present in every cell of the body. Their effect on survival may therefore be mediated by cell types or tissues other than the cancerous breast tissue. These include the various cell types present in the tumor microenvironment or distant tissues that form the pre-metastatic niche. Furthermore, a PPI network specific for healthy breast tissue may not accurately describe the interactions active in transformed cancer cells. In our analysis, we used a generic PPI network. To prevent false positive modules, we complemented the network propagation with an extra filtering step in which we select high-confidence modules based on their association with survival.

Using curated protein interaction networks such as iRefIndex in propagation analyses may cause a subtle type of ascertainment bias: more interactions tend to be known for better studied proteins, which proteins involved in tumor initiation and progression often are. As a result, gene scores may correlate positively with the number of interactions in the protein interaction network. This is the case, for example, when gene scores are based on somatic mutation frequencies in cancer. HotNet2 only controls for this partially, whereas a recent extension to the HotNet2 method provides a more rigorous solution[67]. We tested whether our analysis was vulnerable to this ascertainment bias by calculating the correlation between the gene scores computed by Pascal and the number of interactions recorded by iRefIndex. For all, ER-positive, and ER-negative breast cancer, these correlations were close to zero (Pearson $r^2 = -0.012$, $r^2 = -0.006$, and $r^2 = 0.003$, respectively) showing no evidence of ascertainment bias due to proteins' numbers of recorded interactions.

In summary, our network propagation analysis shows a germline genetic link to breast cancer survival and proposes a mechanism by which multiple loci with small individual effects might influence breast cancer-specific prognosis. Experimental follow-up of the high-confidence GRPMs identified is required to better understand the role of these modules. While we focused on the subset of high-confidence modules, the other modules may also yield new insights if assessed in the context of larger independent datasets. Together the results presented here may feed future hypotheses about the contribution of germline variation to breast cancer survival.

## Methods

**Breast cancer patient data**. We used data from 12 GWASs that together account for 84,457 invasive breast cancer patients with 5413 breast cancer-specific deaths

within 10 years (events). These included 55,701 patients with ER-positive breast cancer (2854 events) and 14,529 patients with ER-negative breast cancer (1724 events), while the ER status was unknown for the remaining 14,227 patients. All patients were females of European ancestry. A summary of the studies with the numbers of patients and events by study is given in Supplementary Table 1. The GWAS sample sets were genotyped using a variety of genotyping arrays, targeting between 200,000 and 900,000 variants across the genome, and subsequently imputed using a common reference (details given below). The majority of patients came from the Breast Cancer Association Consortium (BCAC), which itself comprised 69 studies from across the world that underwent a uniform data harmonization and quality control (data freeze 10). Genotyping in BCAC was performed in two rounds using two different genotyping platforms: iCOGS and OncoArray. In subsequent analyses, we treated these two platforms as different studies. The OncoArray dataset is the largest in BCAC, with higher-quality imputed genotypes compared to the iCOGS data. As an independent dataset, we separated out the entire SEARCH study, comprising 12,381 patients and 1120 events, from the BCAC data. Patients in the SEARCH study were recruited in the United Kingdom. Their genotypes were obtained using either iCOGS or OncoArray (Supplementary Table 2). Participants of all the studies provided written informed consent and studies were approved by local medical ethical committees.

**Genotype data and sample quality control.** Quality checks were performed by the original studies[3,5,68]. Genotypes for all 12 datasets were imputed using a reference panel from the 1000 Genomes Project[69] March 2012 release. Imputation was performed by a two-stage procedure[3] using SHAPEIT[70] for pre-phasing and IMPUTE2[71] for genotype imputation. The genome-wide analyses were performed on ~7.3 million variants that had a minor allele frequency (MAF) > 0.05 and were imputed with imputation quality $r^2 > 0.8$ in at least one of the studies.

**GWAS survival analysis and summary statistics.** The survival analysis was performed for all invasive breast cancer cases combined and for each of the ER status subtypes (ER-positive and ER-negative) individually. A Cox proportional hazards model was fitted to assess the association of the genotype with breast cancer-specific survival. Time to event was calculated from the date of diagnosis. Yet, because patients were recruited at different times before or after diagnosis, time at risk was calculated from the recruitment date (left truncation) in order to avoid possible bias produced by prevalent cases. Follow-up was right censored on the date of death if the patient died from a cause other than breast cancer, the last date the patient was known to be alive if death did not occur, or at 10 years after diagnosis, whatever came first. To control for cryptic population substructure, we adjusted for principal components[3] (for the number of principal components per study, see Supplementary Table 1). Since BCAC-OncoArray and BCAC-iCOGS comprised data from large international cohort studies, the Cox models for these datasets were stratified by country. Separate survival analyses were performed for each of the 12 main studies, after which overall results per variant were obtained by combining the results of all studies with imputation quality $r^2 > 0.8$ for that variant using a fixed-effects meta-analysis. $P$ values were computed using a two-sided Wald test.

**From variant $P$ values to gene scores.** We used the GWAS summary statistics from the survival analysis as input for computing gene scores. To obtain gene scores, we used the Pascal algorithm[12], which combines variant $P$ values while taking into account dependence due to LD structure. The Pascal method implements two gene-level statistics, corresponding to the strongest single association per gene (maximum of chi-squared statistics) or the average of all associations across the gene (sum of chi-squared statistics). After computing both statistics, we tested which one had more power. To this end, we represented the set of $P$ values into a quantile–quantile (QQ)-plot (Supplementary Fig. 6). For all breast cancer cases and for both ER status groups, the QQ-plots suggested that the maximum statistic has more power than the sum statistic. Therefore, of the two gene statistics we chose the maximum of chi-squared statistics for the gene-level statistic.

For the LD reference population used in the gene computation, we created an extended version that included more variants than the default library provided with Pascal. This reference population was based on 503 European genomes from the 1000 Genomes Project (1KG)[69]. For the remaining parameters, we used the default settings. First, only variants with an imputation quality $r^2 > 0.8$ and MAF > 5% in the patient data were considered. Second, the mapping of the variants to genes was based on the Pascal's default 50-kb window size from the start and end of the gene. Finally, when computing gene scores, HLA genes were excluded. After the gene score computation, we obtained 21,815 gene scores for all invasive breast cancer, 21,789 for ER-positive and 21,797 for ER-negative. The slightly different numbers of gene scores between groups are due to the distinct selection of variants, which may have different allele frequencies across groups. The gene scores used in the HotNet2 analysis were obtained by taking the $-\log_{10}$ of the gene $P$ values computed with Pascal.

**Network propagation with HotNet2.** We performed a network propagation analysis using the HotNet2 algorithm[10] and the PPI network iRefIndex[25] applied to the $-\log_{10}$ gene scores obtained from the previous step. For edge removal on the created modules, HotNet2 automatically selects four different values which determine four different edge removal thresholds. The significance test is a two-stage statistical test based on the number and size of the identified modules compared to those found using a permutation test. We used 500 permutations and a minimum network size of 2 for statistical testing. Further details are provided in the original HotNet publication[72,73].

**Construction of PHSs.** To summarize the total prognostic effect of the hereditary variants within the significant GRPMs, we constructed PHSs, using a two-step approach. First, we selected the set of variants that best represented the genetic association of breast cancer survival with each GRPM. This variant selection was performed on the BCAC-OncoArray data, since this was the largest study and had the highest imputation quality. We performed the selection using the *glmnet* R package[74], fitting a Lasso (alpha = 1) model with tenfold cross-validation to tune the sparsity penalty and the same selection of input variants as used for the computation of the Pascal gene scores, that is, picking those variants with MAF > 5% and within a 50-kb window around the start and end of the gene. With the set of germline variants selected using the Lasso procedure (Supplementary Table 3), we fitted a Cox model to estimate unpenalized coefficients and extracted their effect size estimates to compute a PHS per GRPM, which characterized the whole set of variants for the specific module in a unique score. For a set of selected variants {1, …,$n$}, the PHS is defined as in (1):

$$PHS = \sum_{i=1}^{n} X_i \beta_i \tag{1}$$

where $X_i$ is the genotype for the $i$th variant and $\beta_i$ its associated coefficient.

**Identification of high-confidence GRPMs.** We obtained a selection of high-confidence GRPMs from among all modules identified using HotNet2 by testing the association of each module's PHS in two datasets. The first dataset was the BCAC-OncoArray data minus the SEARCH data component of BCAC, i.e., the same data on which the PHS was derived, which was also a subset of the data used in the HotNet2 analysis. The second dataset consisted of the SEARCH study, which was held out of the BCAC data to serve as a truly independent set. Only GRPMs that had a PHS significantly associated ($P < 0.05$) with breast cancer-specific survival in both the BCAC-OncoArray and the independent SEARCH data were considered high-confidence GRPMs and kept for further analysis. To test the association of a PHS with prognosis, we fitted a Cox model to the PHS, adjusted for the first two genetic principal components and stratified by country. We then calculated a one-sided $P$ value for the association of the PHS covariate with survival, taking advantage of the fact that the direction of association of the PHS is predefined, i.e., lower PHS means better survival. For the BCAC OncoArray data, the $P$ value was corrected for multiple testing using Bonferroni correction based on the number of modules tested. The independent SEARCH data comprised two subsets using either OncoArray or iCOGS data. We analyzed these two subsets separately and then combined the results of both groups using a fixed-effect meta-analysis.

**Functional enrichment analysis of GRPM members.** Using the Cytoscape version 3.4.0 software[75], we extended the GRPMs by adding the first direct neighboring genes in the Mentha[76] human PPI network. With the extension of the GRPMs, we obtained bigger modules placed in a functional context. We then used the Cytoscape app ClueGO[77]. ClueGO uses kappa statistics to group the elements of the network and creates organized pathway categories based on the integrated pathway annotation. We based the analysis on human Reactome[28] pathways, a Kappa Score Threshold of 0.4, and Bonferroni correction for the computed enrichment $P$ values. For the visualization, we selected the fusion feature that groups pathways according to overlapping genes to facilitate interpretation of the results. We selected pathways with a $P$ value <0.05.

**Downstream functional enrichment.** In order to add biological and functional interpretation to the GRPMs, we looked for associations between the modules' PHSs and the expression patterns of potential downstream genes (Fig. 1e). From TCGA[26] library, we extracted matched RNA-seq and genotype data of female breast cancer patients of European ancestry. This resulted in 118 patients with ER-negative breast cancer and 440 patients with ER-positive breast cancer. For each GRPM, we computed the previously obtained PHS for the subset of TCGA patients with a tumor matching the subtype for which the GRPM was found. Next, we aimed to quantify the downstream transcriptional effect of the GRPM on the expression of every individual gene. To do so, we computed the Pearson correlation between the GRPM's PHS and the RNA expression of each gene. Finally, we performed GSEA[27] to test for enrichment of biological pathways among the highly correlating genes. We used an annotation set of Reactome pathways[28] and MSigDB[29] Hallmark gene sets to perform the pre-ranked GSEA. We visualized the Reactome results with the EnrichmentMap[78] Cytoscape app. Only biological processes with $P$ value <0.001 and FDR < 0.05 were considered as significantly enriched.

**Ethical Approval**. The study was performed in accordance with the Declaration of Helsinki. All individual studies, from which data was used, were approved by the appropriate medical ethical committees and/or institutional review boards. All study participants provided informed consent.

**Reporting summary**. Further information on research design is available in the Nature Research Reporting Summary linked to this article.

## Data availability

All 10-year breast cancer-specific survival summary estimates are available via the BCAC website (http://bcac.ccge.medschl.cam.ac.uk/bcacdata/). Individual patient data will not be made publicly available without request due to restraints imposed by the ethics committees of individual studies. Formal request can be made via the Data Access Coordination Committee (DACC) of BCAC (http://bcac.ccge.medschl.cam.ac.uk/). A subset of the data that supports the findings of this analysis is available at https://portal.gdc.cancer.gov/ (accession number phs000178).

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

## Acknowledgements

BCAC: We thank all the individuals who took part in these studies and all the researchers, clinicians, technicians, and administrative staff who have enabled this work to be carried out. We acknowledge all contributors to the COGS and OncoArray study design, chip design, genotyping, and genotype analyses. ABCFS: Maggie Angelakos, Judi Maskiell, Gillian Dite. ABCS: Frans Hogervorst, Sten Cornelissen and Annegien Broeks. ABCTB Investigators: Rosemary Balleine, Robert Baxter, Stephen Braye, Jane Carpenter, Jane Dahlstrom, John Forbes, Soon Lee, Debbie Marsh, Adrienne Morey, Nirmala Pathmanathan, Rodney Scott, Allan Spigelman, Nicholas Wilcken, Desmond Yip. **BBCS:** Eileen Williams, Elaine Ryder-Mills, Kara Sargus. BCINIS: Dr. K. Landsman, Dr. N. Gronich, Dr. A. Flugelman, Dr. W. Saliba, Dr. E. Liani, Dr. I. Cohen, Dr. S. Kalet, Dr. V. Friedman, Dr. O. Barnet. **BIGGS:** Niall McInerney, Gabrielle Colleran, Andrew Rowan, Angela Jones. BREOGAN: Manuela Gago-Dominguez, Jose Esteban Castelao, Angel Carracedo, Victor Muñoz Garzón, Alejandro Novo Domínguez, Maria Elena Martinez, Sara Miranda Ponte, Carmen Redondo Marey, Maite Peña Fernández, Manuel Enguix Castelo, Maria Torres, Manuel Calaza, José Antúnez, Máximo Fraga; Joaquín González-Carreró and the Department of Pathology and Biobank of University Hospital Complex of Vigo, Instituto de Investigacion Biomedica Galicia Sur, SERGAS. BSUCH: Peter Bugert, Medical Faculty Mannheim. CCGP: Styliani Apostolaki, Anna Margiolaki, Georgios Nintos, Maria Perraki, Georgia Saloustrou, Georgia Sevastaki, Konstantinos Pompodakis. CGPS: Dorthe Uldall Andersen, Maria Birna Arnadottir, Anne Bank, Dorthe Kjeldgård Hansen, and the Danish Cancer Biobank. CNIO-BCS: Guillermo Pita, Charo Alonso, Nuria Álvarez, Pilar Zamora, and Primitiva Menendez. CPS-II: Centers for Disease Control and Prevention National Program of Cancer Registries. The National Cancer Institute Surveillance Epidemiology, and End Results program. CTS: Leslie Bernstein, Susan Neuhausen, James Lacey, Sophia Wang, Huiyan Ma, and Jessica Clague DeHart. Dennis Deapen, Rich Pinder, and Eunjung Lee, Pam Horn-Ross, Peggy Reynolds, Christina Clarke Dur and David Nelson, Hoda Anton-Culver, Argyrios Ziogas, and Hannah Park and Fred Schumacher. DIETCOMPLYF: charity Against Breast Cancer (Registered Charity Number 1121258) and the NCRN. Participants and the investigators of EPIC (European Prospective Investigation into Cancer and Nutrition). ESTHER: Hartwig Ziegler, Sonja Wolf, Volker Hermann, Christa Stegmaier, Katja Butterbach. FHRISK: NIHR for funding. GC-HBOC: Stefanie Engert, Heide Hellebrand, Sandra Kröber and LIFE. Markus Loeffler, Joachim Thiery, Matthias Nüchter, Ronny Baber. GENICA: Dr. Margarete Fischer-Bosch [HB, Wing-Yee Lo], German Cancer Consortium (DKTK), and German Cancer Research Center (DKFZ) [HB], gefördert durch die Deutsche Forschungsgemeinschaft (DFG) im Rahmen der Exzellenzstrategie des Bundes und der Länder - EXC 2180 - 390900677 [HB], Evangelische Kliniken Bonn gGmbH, Johanniter Krankenhaus, [Yon-Dschun Ko, Christian Baisch], University of Bonn, Germany [Hans-Peter Fischer], Deutsches Krebsforschungszentrum (DKFZ), Heidelberg, Germany [UH], Institute for Prevention and Occupational Medicine of the German Social Accident Insurance, Institute of the Ruhr University Bochum (IPA), Bochum, Germany [Thomas Brüning, Beate Pesch, Sylvia Rabstein, Anne Lotz]; University Medical Center Hamburg-Eppendorf, Germany [Volker Harth]. HABCS: Michael Bremer. HEBCS: Rainer Fagerholm, Kirsimari Aaltonen, Karl von Smitten, Irja Erkkilä. HUBCS: Shamil Gantsev. KARMA and SASBAC: Swedish Medical Research Counsel. KBCP: Eija Myöhänen, Helena Kemiläinen. kConFab/AOCS: Eveline Niedermayr, Family Cancer Clinics and the Clinical Follow Up Study (received funding from the NHMRC, the National Breast Cancer Foundation, Cancer Australia, and the National Institute of Health (USA)). LMBC: Gilian Peuteman, Thomas Van Brussel, Evy-Vanderheyden and Kathleen Corthouts. MARIE: Petra Seibold, Judith Heinz, Nadia Obi, Sabine Behrens, Ursula Eilber, Muhabbet Celik and Til Olchers. MBCSG: Paolo Radice, Jacopo Azzollini, Bernardo Bonanni, Bernard Peissel, Roberto Villa, Giulia Cagnoli, Irene Feroce, and Cogentech Cancer Genetic Test Laboratory. NBCS: Kristine K. Sahlberg (PhD), Lars Ottestad (MD), Rolf Kåresen (Prof. Em.) Dr. Ellen Schlichting (MD), Marit Muri Holmen (MD), Toril Sauer (MD), Vilde Haakensen (MD), Olav Engebråten (MD), Bjørn Naume (MD), Alexander Fosså (MD), Cecile E. Kiserud (MD), Kristin V. Reinertsen (MD), Åslaug Helland (MD), Margit Riis (MD), Jürgen Geisler (MD) and OSBREAC. NHS/NHS2: AL, AZ, AR, CA, CO, CT, DE, FL, GA, ID, IL, IN, IA, KY, LA, ME, MD, MA, MI, NE, NH, NJ, NY, NC, ND, OH, OK, OR, PA, RI, SC, TN, TX, VA, WA, WY. OBCS: Arja Jukkola-Vuorinen, Mervi Grip, Saila Kauppila, Meeri Otsukka, Leena Keskitalo and Kari Mononen. OFBCR: Teresa Selander, Nayana Weerasooriya. ORIGO: E. Krol-Warmerdam, and J. Blom. PBCS: Louise Brinton, Mark Sherman, Neonila Szeszenia-Dabrowska, Beata Peplonska, Witold Zatonski, Pei Chao, Michael Stagner. The ethical approval for the POSH study is MREC /00/6/69, UKCRN ID: 1137. Experimental Cancer Medicine Centre (ECMC) supported Faculty of Medicine Tissue Bank and the Faculty of Medicine DNA Banking resource. PREFACE: Sonja Oeser and Silke Landrith. PROCAS: NIHR for funding. RBCS: Petra Bos, Jannet Blom, Ellen Crepin, Elisabeth Huijskens, Anja Kromwijk-Nieuwlaat, Annette Heemskerk, the Erasmus MC Family Cancer Clinic. SBCS: Sue Higham, Helen Cramp, Dan Connley, Ian Brock, Sabapathy Balasubramanian and Malcolm W.R. Reed. We thank the SEARCH and EPIC teams. SKKDKFZS: SUCCESS Study teams in Munich, Duessldorf, Erlangen and Ulm. SZBCS: Ewa Putresza. UCIBCS: Irene Masunaka. UKBGS: Breast Cancer Now and the Institute of Cancer Research and NHS funding to the Royal Marsden/ICR NIHR Biomedical Research Centre. WHI: investigators and staff for their dedication. BCAC is funded by Cancer Research UK [C1287/A16563, C1287/A10118], the European Union's Horizon 2020 Research and Innovation Programme (grant numbers 634935 and 633784 for BRIDGES and B-CAST respectively), and by the European Community's Seventh Framework Programme under grant agreement number 223175 (grant number HEALTH-F2-2009-223175) (COGS). The EU Horizon 2020 Research and Innovation Programme funding source had no role in study design, data collection, data analysis, data interpretation or writing of the report. Genotyping of the OncoArray was funded by the NIH Grant U19 CA148065, and Cancer UK Grant C1287/A16563 and the PERSPECTIVE project supported by the Government of Canada through Genome Canada and the Canadian Institutes of Health Research (grant GPH-129344) and, the Ministère de l'Économie, Science et Innovation du Québec through Genome Québec and the

PSRSIIRI-701 grant, and the Quebec Breast Cancer Foundation. Funding for the iCOGS infrastructure came from: the European Community's Seventh Framework Programme under grant agreement n° 223175 (HEALTH-F2-2009-223175) (COGS), Cancer Research UK (C1287/A10118, C1287/A10710, C12292/A11174, C1281/A12014, C5047/A8384, C5047/A15007, C5047/A10692, C8197/A16565), the National Institutes of Health (CA128978) and Post-Cancer GWAS initiative (1U19 CA148537, 1U19 CA148065 and 1U19 CA148112 - the GAME-ON initiative), the Department of Defence (W81XWH-10-1-0341), the Canadian Institutes of Health Research (CIHR) for the CIHR Team in Familial Risks of Breast Cancer, and Komen Foundation for the Cure, the Breast Cancer Research Foundation, and the Ovarian Cancer Research Fund. The DRIVE Consortium was funded by U19 CA148065. ABCFS was supported by grant UM1 CA164920 from the National Cancer Institute (USA). The content of this manuscript does not necessarily reflect the views or policies of the National Cancer Institute or any of the collaborating centers in the in the Breast Cancer Family Registry (BCFR), nor does mention of trade names, commercial products, or organizations imply endorsement by the USA Government or the BCFR. The ABCFS was also supported by the National Health and Medical Research Council of Australia, the New South Wales Cancer Council, the Victorian Health Promotion Foundation (Australia) and the Victorian Breast Cancer Research Consortium. J.L.H. is a National Health and Medical Research Council (NHMRC) Senior Principal Research Fellow. M.C.S. is a NHMRC Senior Research Fellow. The ABCS study was supported by the Dutch Cancer Society [grants NKI 2007-3839; 2009-4363; 2015-7632]. The ABCTB was supported by the National Health and Medical Research Council of Australia, The Cancer Institute NSW and the National Breast Cancer Foundation. The work of the BBCC was partly funded by ELAN-Fond of the University Hospital of Erlangen. The BBCS is funded by Cancer Research UK and Breast Cancer Now and acknowledges NHS funding to the NIHR Biomedical Research Centre, and the National Cancer Research Network (NCRN). For the BCFR-NY, BCFR-PA, BCFR-UT this work was supported by grant UM1 CA164920 from the National Cancer Institute. For BIGGS, ES is supported by NIHR Comprehensive Biomedical Research Centre, Guy's & St. Thomas' NHS Foundation Trust in partnership with King's College London, United Kingdom. IT is supported by the Oxford Biomedical Research Centre. The BREOGAN is funded by Acción Estratégica de Salud del Instituto de Salud Carlos III FIS PI12/02125/Cofinanciado FEDER; Acción Estratégica de Salud del Instituto de Salud Carlos III FIS PI17/00918/Cofinanciado FEDER; Acción Estratégica de Salud del Instituto de Salud Carlos III FIS Intrasalud (PI13/01136); Programa Grupos Emergentes, Cancer Genetics Unit, Instituto de Investigacion Biomedica Galicia Sur. Xerencia de Xestion Integrada de Vigo-SERGAS, Instituto de Salud Carlos III, Spain; Grant 10CSA012E, Consellería de Industria Programa Sectorial de Investigación Aplicada, PEME I + D e I + D Suma del Plan Gallego de Investigación, Desarrollo e Innovación Tecnológica de la Consellería de Industria de la Xunta de Galicia, Spain; Grant EC11-192. Fomento de la Investigación Clínica Independiente, Ministerio de Sanidad, Servicios Sociales e Igualdad, Spain; and Grant FEDER-Innterconecta. Ministerio de Economia y Competitividad, Xunta de Galicia, Spain. The BSUCH study was supported by the Dietmar-Hopp Foundation, the Helmholtz Society and the German Cancer Research Center (DKFZ). CCGP is supported by funding from the University of Crete. The CECILE study was supported by Fondation de France, Institut National du Cancer (INCa), Ligue Nationale contre le Cancer, Agence Nationale de Sécurité Sanitaire, de l'Alimentation, de l'Environnement et du Travail (ANSES), Agence Nationale de la Recherche (ANR). The CGPS was supported by the Chief Physician Johan Boserup and Lise Boserup Fund, the Danish Medical Research Council, and Herlev and Gentofte Hospital. The CNIO-BCS was supported by the Instituto de Salud Carlos III, the Red Temática de Investigación Cooperativa en Cáncer and grants from the Asociación Española Contra el Cáncer and the Fondo de Investigación Sanitario (PI11/00923 and PI12/00070). The American Cancer Society funds the creation, maintenance, and updating of the CPS-II cohort. The CTS was initially supported by the California Breast Cancer Act of 1993 and the California Breast Cancer Research Fund (contract 97-10500) and is currently funded through the National Institutes of Health (R01 CA77398, UM1 CA164917, and U01 CA199277). Collection of cancer incidence data was supported by the California Department of Public Health as part of the statewide cancer reporting program mandated by California Health and Safety Code Section 103885. The University of Westminster curates the DietCompLyf database funded by Against Breast Cancer Registered Charity No. 1121258 and the NCRN. The coordination of EPIC is financially supported by the European Commission (DG-SANCO) and the International Agency for Research on Cancer. The national cohorts are supported by: Ligue Contre le Cancer, Institut Gustave Roussy, Mutuelle Générale de l'Education Nationale, Institut National de la Santé et de la Recherche Médicale (INSERM) (France); German Cancer Aid, German Cancer Research Center (DKFZ), Federal Ministry of Education and Research (BMBF) (Germany); the Hellenic Health Foundation, the Stavros Niarchos Foundation (Greece); Associazione Italiana per la Ricerca sul Cancro-AIRC-Italy and National Research Council (Italy); Dutch Ministry of Public Health, Welfare and Sports (VWS), Netherlands Cancer Registry (NKR), LK Research Funds, Dutch Prevention Funds, Dutch ZON (Zorg Onderzoek Nederland), World Cancer Research Fund (WCRF), Statistics Netherlands (The Netherlands); Health Research Fund (FIS), PI13/00061 to Granada, PI13/01162 to EPIC-Murcia, Regional Governments of Andalucía, Asturias, Basque Country, Murcia and Navarra, ISCIII RETIC (RD06/0020) (Spain); Cancer Research UK (14136 to EPIC-Norfolk; C570/A16491 and C8221/A19170 to EPIC-Oxford), Medical Research Council (1000143 to EPIC-Norfolk, MR/M012190/1 to EPIC-Oxford) (United Kingdom). The ESTHER study was supported by a grant from the Baden Württemberg Ministry of Science, Research and Arts. Additional cases were recruited in the context of the VERDI study, which was supported by a grant from the German Cancer Aid (Deutsche Krebshilfe). FHRISK is funded from NIHR grant PGfAR 0707-10031. Prof D Gareth Evans is supported by the NIHR Manchester Biomedical Research Centre (IS-BRC-1215-20007). The GC-HBOC is supported by the German Cancer Aid (grant no 110837, coordinator: Rita K. Schmutzler, Cologne). This work was also funded by the European Regional Development Fund and Free State of Saxony, Germany (LIFE - Leipzig Research Centre for Civilization Diseases, project numbers 713-241202, 713-241202, 14505/2470, 14575/2470). The GENICA was funded by the Federal Ministry of Education and Research (BMBF) Germany grants 01KW9975/5, 01KW9976/8, 01KW9977/0, and 01KW0114, the Robert Bosch Foundation, Stuttgart, Deutsches Krebsforschungszentrum (DKFZ), Heidelberg, the Institute for Prevention and Occupational Medicine of the German Social Accident Insurance, Institute of the Ruhr University Bochum (IPA), Bochum, as well as the Department of Internal Medicine, Evangelische Kliniken Bonn gGmbH, Johanniter Krankenhaus, Bonn, Germany. The GESBC was supported by the Deutsche Krebshilfe e. V. [70492] and the German Cancer Research Center (DKFZ). The HABCS study was supported by the Claudia von Schilling Foundation for Breast Cancer Research, by the Lower Saxonian Cancer Society, and by the Rudolf Bartling Foundation. The HEBCS was financially supported by the Helsinki University Central Hospital Research Fund, Academy of Finland (266528), the Finnish Cancer Society, and the Sigrid Juselius Foundation. The HUBCS was supported by a grant from the German Federal Ministry of Research and Education (RUS08/017), and by the Russian Foundation for Basic Research and the Federal Agency for Scientific Organizations for support the Bioresource collections and RFBR grants 14-04-97088, 17-29-06014 and 17-44-020498. Financial support for KARBAC was provided through the regional agreement on medical training and clinical research (ALF) between Stockholm County Council and Karolinska Institutet, the Swedish Cancer Society, The Gustav V Jubilee foundation and Bert von Kantzows foundation. The KARMA study was supported by Märit and Hans Rausings Initiative Against Breast Cancer. The KBCP was financially supported by the special Government Funding (EVO) of Kuopio University Hospital grants, Cancer Fund of North Savo, the Finnish Cancer Organizations, and by the strategic funding of the University of Eastern Finland. kConFab is supported by a grant from the National Breast Cancer Foundation, and previously by the National Health and Medical Research Council (NHMRC), the Queensland Cancer Fund, the Cancer Councils of New South Wales, Victoria, Tasmania and South Australia, and the Cancer Foundation of Western Australia. LMBC is supported by the "Stichting tegen Kanker." The MARIE study was supported by the Deutsche Krebshilfe e.V. [70-2892-BR I, 106332, 108253, 108419, 110826, 110828], the Hamburg Cancer Society, the German Cancer Research Center (DKFZ) and the Federal Ministry of Education and Research (BMBF) Germany [01KH0402]. MBCSG is supported by grants from the Italian Association for Cancer Research (AIRC) and by funds from the Italian citizens who allocated the 5/1000 share of their tax payment in support of the Fondazione IRCCS Istituto Nazionale Tumori, according to Italian laws (INT-Institutional strategic projects "5x1000"). The MCBCS was supported by the NIH grants CA192393, CA116167, CA176785 an NIH Specialized Program of Research Excellence (SPORE) in Breast Cancer [CA116201], and the Breast Cancer Research Foundation and a generous.pngt from the David F. and Margaret T. Grohne Family Foundation. MCCS cohort recruitment was funded by VicHealth and Cancer Council Victoria. The MCCS was further supported by Australian NHMRC grants 209057 and 396414, and by infrastructure provided by Cancer Council Victoria. Cases and their vital status were ascertained through the Victorian Cancer Registry (VCR) and the Australian Institute of Health and Welfare (AIHW), including the National Death Index and the Australian Cancer Database. The MEC was supported by NIH grants CA63464, CA54281, CA098758, CA132839 and CA164973. The MISS study is supported by funding from ERC-2011-294576 Advanced grant, Swedish Cancer Society, Swedish Research Council, Local hospital funds, Berta Kamprad Foundation, Gunnar Nilsson. The MMHS study was supported by NIH grants CA97396, CA128931, CA116201, CA140286 and CA177150. The NBCS has received funding from the K.G. Jebsen Centre for Breast Cancer Research; the Research Council of Norway grant 193387/V50 (to A-L Børresen-Dale and V.N. Kristensen) and grant 193387/H10 (to A-L Børresen-Dale and V.N. Kristensen), South Eastern Norway Health Authority (grant 39346 to A-L Børresen-Dale) and the Norwegian Cancer Society (to A-L Børresen-Dale and V.N. Kristensen). The NC-BCFR and OFBCR were supported by grant UM1 CA164920 from the National Cancer Institute (USA). The NCBCS was funded by Komen Foundation, the National Cancer Institute (P50 CA058223, U54 CA156733, U01 CA179715), and the North Carolina University Cancer Research Fund. The NHS was supported by NIH grants P01 CA87969, UM1 CA186107, and U19 CA148065. The NHS2 was supported by NIH grants UM1 CA176726 and U19 CA148065. The OBCS was supported by research grants from the Finnish Cancer Foundation, the Academy of Finland (grant number 250083, 122715 and Center of Excellence grant number 251314), the Finnish Cancer Foundation, the Sigrid Juselius Foundation, the University of Oulu, the University of Oulu Support Foundation and the special Governmental EVO funds for Oulu University Hospital-based research activities. The ORIGO study was supported by the Dutch Cancer Society (RUL 1997-1505) and the Biobanking and Biomolecular Resources Research Infrastructure (BBMRI-NL CP16). The PBCS was funded by Intramural Research Funds of the National Cancer Institute, Department of Health and Human Services, USA. Genotyping for PLCO was supported by the Intramural Research Program of the National Institutes of Health, NCI, Division of Cancer Epidemiology and Genetics. The PLCO is supported by the

Intramural Research Program of the Division of Cancer Epidemiology and Genetics and supported by contracts from the Division of Cancer Prevention, National Cancer Institute, National Institutes of Health. The POSH study is funded by Cancer Research UK (grants C1275/A11699, C1275/C22524, C1275/A19187, C1275/A15956 and Breast Cancer Campaign 2010PR62, 2013PR044. PROCAS is funded from NIHR grant PGfAR 0707-10031. PROCAS is funded from NIHR grant PGfAR 0707-10031. The RBCS was funded by the Dutch Cancer Society (DDHK 2004-3124, DDHK 2009-4318). The SASBAC study was supported by funding from the Agency for Science, Technology and Research of Singapore (A*STAR), the US National Institute of Health (NIH) and the Susan G. Komen Breast Cancer Foundation. The SBCS was supported by Sheffield Experimental Cancer Medicine Centre and Breast Cancer Now Tissue Bank. SEARCH is funded by Cancer Research UK [C490/A10124, C490/A16561] and supported by the UK National Institute for Health Research Biomedical Research Centre at the University of Cambridge. The University of Cambridge has received salary support for PDPP from the NHS in the East of England through the Clinical Academic Reserve. SKKDKFZS is supported by the DKFZ. The SMC is funded by the Swedish Cancer Foundation and the Swedish Research Council (SIMPLER, VR 2017-00644). The SZBCS was supported by Grant PBZ_KBN_122/P05/2004. The UCIBCS component of this research was supported by the NIH [CA58860, CA92044] and the Lon V Smith Foundation [LVS39420]. The UKBGS is funded by Breast Cancer Now and the Institute of Cancer Research (ICR), London. ICR acknowledges NHS funding to the NIHR Biomedical Research Centre. The USRT Study was funded by Intramural Research Funds of the National Cancer Institute, Department of Health and Human Services, USA. The WHI program is funded by the National Heart, Lung, and Blood Institute, the US National Institutes of Health and the US Department of Health and Human Services (HHSN268201100046C, HHSN268201100001C, HHSN268201100002C, HHSN268201100003C, HHSN268201100004C, and HHSN271201100004C). This work was also funded by NCI U19 CA148065-01.

## Author contributions

M.K.S. and S.C. conceived the study. M.E.G. performed the data analyses. M.K.S., S.C. and M.E.G. were involved in the interpretation of the data. S.C. provided statistical and computational support for the data analyses. R.K., Q.W., M.K.B. and J.D. provided database support. M.E.G., M.K.S. and S.C. wrote the manuscript. All authors contributed data from their own studies, helped revise the manuscript, and approved the final version. All authors consented to this publication.

## Competing interests

A.Ashworth is a cofounder of Tango Therapeutics, Azkarra Therapeutics, and Ovibio, is an advisor for Gladiator, Prolynx, Bluestar, Earli and Genentech, reports receiving commercial research grants from AstraZeneca and SPARC, and has ownership interest in patents on the use of PARP inhibitors, held jointly with AstraZeneca. The remaining authors declare no competing interests.

## Additional information

Maria Escala-Garcia[1], Jean Abraham[2,3,4], Irene L. Andrulis[5,6], Hoda Anton-Culver[7], Volker Arndt[8], Alan Ashworth[9], Paul L. Auer[10,11], Päivi Auvinen[12,13,14], Matthias W. Beckmann[15], Jonathan Beesley[16], Sabine Behrens[17], Javier Benitez[18,19], Marina Bermisheva[20], Carl Blomqvist[21,22], William Blot[23,24], Natalia V. Bogdanova[25,26,27], Stig E. Bojesen[28,29,30], Manjeet K. Bolla[31], Anne-Lise Børresen-Dale[32,33], Hiltrud Brauch[34,35,36], Hermann Brenner[8,36,37], Sara Y. Brucker[38], Barbara Burwinkel[39,40], Carlos Caldas[41,42], Federico Canzian[43], Jenny Chang-Claude[17,44], Stephen J. Chanock[45], Suet-Feung Chin[46], Christine L. Clarke[47], Fergus J. Couch[48], Angela Cox[49], Simon S. Cross[50], Kamila Czene[51], Mary B. Daly[52], Joe Dennis[31], Peter Devilee[53,54], Janet A. Dunn[55], Alison M. Dunning[2], Miriam Dwek[56], Helena M. Earl[4,57], Diana M. Eccles[58], A. Heather Eliassen[59,60], Carolina Ellberg[61], D. Gareth Evans[62,63,64], Peter A. Fasching[15,65], Jonine Figueroa[45,66,67], Henrik Flyger[68], Manuela Gago-Dominguez[69,70], Susan M. Gapstur[71], Montserrat García-Closas[45,72], José A. García-Sáenz[73], Mia M. Gaudet[71], Angela George[74], Graham G. Giles[75,76,77], David E. Goldgar[78], Anna González-Neira[18], Mervi Grip[79], Pascal Guénel[80], Qi Guo[81], Christopher A. Haiman[82], Niclas Håkansson[83], Ute Hamann[84], Patricia A. Harrington[2], Louise Hiller[55], Maartje J. Hooning[85], John L. Hopper[76], Anthony Howell[86], Chiun-Sheng Huang[87], Guanmengqian Huang[84], David J. Hunter[60,88,89], Anna Jakubowska[90,91], Esther M. John[92], Rudolf Kaaks[17], Pooja Middha Kapoor[17,93], Renske Keeman[1], Cari M. Kitahara[94], Linetta B. Koppert[95], Peter Kraft[60,88], Vessela N. Kristensen[32,33], Diether Lambrechts[96,97], Loic Le Marchand[98], Flavio Lejbkowicz[99], Annika Lindblom[100,101], Jan Lubiński[90], Arto Mannermaa[14,102,103], Mehdi Manoochehri[84], Siranoush Manoukian[104], Sara Margolin[105,106],

Maria Elena Martinez[70,107], Tabea Maurer[44], Dimitrios Mavroudis[108], Alfons Meindl[109], Roger L. Milne[75,76,110], Anna Marie Mulligan[111,112], Susan L. Neuhausen[113], Heli Nevanlinna[114], William G. Newman[62,63], Andrew F. Olshan[115], Janet E. Olson[116], Håkan Olsson[61], Nick Orr[117], Paolo Peterlongo[118], Christos Petridis[119], Ross L. Prentice[10], Nadege Presneau[56], Kevin Punie[120], Dhanya Ramachandran[26], Gad Rennert[99], Atocha Romero[121], Mythily Sachchithananthan[47], Emmanouil Saloustros[122], Elinor J. Sawyer[119], Rita K. Schmutzler[123,124], Lukas Schwentner[125], Christopher Scott[116], Jacques Simard[126], Christof Sohn[127], Melissa C. Southey[110,128], Anthony J. Swerdlow[74,129], Rulla M. Tamimi[59,60,88], William J. Tapper[130], Manuel R. Teixeira[131,132], Mary Beth Terry[133], Heather Thorne[134,135], Rob A.E.M. Tollenaar[136], Ian Tomlinson[137,138], Melissa A. Troester[115], Thérèse Truong[80], Clare Turnbull[74], Celine M. Vachon[116], Lizet E. van der Kolk[139], Qin Wang[31], Robert Winqvist[140,141], Alicja Wolk[83,142], Xiaohong R. Yang[45], Argyrios Ziogas[7], Paul D.P. Pharoah[2,31], Per Hall[51,105], Lodewyk F.A. Wessels[143,144], Georgia Chenevix-Trench[16], Gary D. Bader[6,145], Thilo Dörk[26], Douglas F. Easton[2,31], Sander Canisius[1,143,147]* & Marjanka K. Schmidt[1,146,147]*

[1]Division of Molecular Pathology, The Netherlands Cancer Institute - Antoni van Leeuwenhoek Hospital, Amsterdam, The Netherlands. [2]Department of Oncology, Centre for Cancer Genetic Epidemiology, University of Cambridge, Cambridge, UK. [3]Cambridge Experimental Cancer Medicine Centre, Cambridge, UK. [4]Cambridge Breast Unit and NIHR Cambridge Biomedical Research Centre, University of Cambridge NHS Foundation Hospitals, Cambridge, UK. [5]Fred A. Litwin Center for Cancer Genetics, Lunenfeld-Tanenbaum Research Institute of Mount Sinai Hospital, Toronto, ON, Canada. [6]Department of Molecular Genetics, University of Toronto, Toronto, Canada. [7]Department of Epidemiology, Genetic Epidemiology Research Institute, University of California Irvine, Irvine, CA, USA. [8]Division of Clinical Epidemiology and Aging Research, German Cancer Research Center (DKFZ), Heidelberg, Germany. [9]UCSF Helen Diller Family Comprehensive Cancer Center, University of California San Francisco, San Francisco, CA, USA. [10]Cancer Prevention Program, Fred Hutchinson Cancer Research Center, Seattle, WA, USA. [11]Zilber School of Public Health, University of Wisconsin-Milwaukee, Milwaukee, WI, USA. [12]Cancer Center, Kuopio University Hospital, Kuopio, Finland. [13]Institute of Clinical Medicine, Oncology, University of Eastern Finland, Kuopio, Finland. [14]Translational Cancer Research Area, University of Eastern Finland, Kuopio, Finland. [15]Department of Gynecology and Obstetrics, Comprehensive Cancer Center ER-EMN, University Hospital Erlangen, Friedrich-Alexander-University Erlangen-Nuremberg, Erlangen, Germany. [16]Department of Genetics and Computational Biology, QIMR Berghofer Medical Research Institute, Brisbane, QLD, Australia. [17]Division of Cancer Epidemiology, German Cancer Research Center (DKFZ), Heidelberg, Germany. [18]Human Cancer Genetics Programme, Spanish National Cancer Research Centre (CNIO), Madrid, Spain. [19]Biomedical Network on Rare Diseases (CIBERER), Madrid, Spain. [20]Institute of Biochemistry and Genetics, Ufa Scientific Center of Russian Academy of Sciences, Ufa, Russia. [21]Department of Oncology, Helsinki University Hospital, University of Helsinki, Helsinki, Finland. [22]Department of Oncology, Örebro University Hospital, Örebro, Sweden. [23]Division of Epidemiology, Department of Medicine, Vanderbilt Epidemiology Center, Vanderbilt-Ingram Cancer Center, Vanderbilt University School of Medicine, Nashville, TN, USA. [24]International Epidemiology Institute, Rockville, MD, USA. [25]Department of Radiation Oncology, Hannover Medical School, Hannover, Germany. [26]Gynaecology Research Unit, Hannover Medical School, Hannover, Germany. [27]N.N. Alexandrov Research Institute of Oncology and Medical Radiology, Minsk, Belarus. [28]Copenhagen General Population Study, Herlev and Gentofte Hospital, Copenhagen University Hospital, Herlev, Denmark. [29]Department of Clinical Biochemistry, Herlev and Gentofte Hospital, Copenhagen University Hospital, Herlev, Denmark. [30]Faculty of Health and Medical Sciences, University of Copenhagen, Copenhagen, Denmark. [31]Department of Public Health and Primary Care, Centre for Cancer Genetic Epidemiology, University of Cambridge, Cambridge, UK. [32]Department of Cancer Genetics, Institute for Cancer Research, Oslo University Hospital-Radiumhospitalet, Oslo, Norway. [33]Institute of Clinical Medicine, Faculty of Medicine, University of Oslo, Oslo, Norway. [34]Dr. Margarete Fischer-Bosch-Institute of Clinical Pharmacology, Stuttgart, Germany. [35]iFIT-Cluster of Excellence, University of Tuebingen, Tuebingen, Germany. [36]German Cancer Research Center (DKFZ), German Cancer Consortium (DKTK), Heidelberg, Germany. [37]Division of Preventive Oncology, German Cancer Research Center (DKFZ) and National Center for Tumor Diseases (NCT), Heidelberg, Germany. [38]Department of Gynecology and Obstetrics, University of Tübingen, Tübingen, Germany. [39]Molecular Epidemiology Group, C080, German Cancer Research Center (DKFZ), Heidelberg, Germany. [40]Molecular Biology of Breast Cancer, University Womens Clinic Heidelberg, University of Heidelberg, Heidelberg, Germany. [41]Cancer Research UK Cambridge Institute, Department of Oncology, Li Ka Shing Centre, University of Cambridge, Cambridge, UK. [42]Breast Cancer Programme, CRUK Cambridge Cancer Centre and NIHR Biomedical Research Centre, Cambridge University Hospitals NHS Foundation Trust, Cambridge, UK. [43]Genomic Epidemiology Group, German Cancer Research Center (DKFZ), Heidelberg, Germany. [44]Cancer Epidemiology Group, University Cancer Center Hamburg (UCCH), University Medical Center Hamburg-Eppendorf, Hamburg, Germany. [45]Division of Cancer Epidemiology and Genetics, Department of Health and Human Services, National Cancer Institute, National Institutes of Health, Bethesda, MD, USA. [46]Cancer Research UK Cambridge Institute, University of Cambridge, Cambridge, UK. [47]Westmead Institute for Medical Research, University of Sydney, Sydney, NSW, Australia. [48]Department of Laboratory Medicine and Pathology, Mayo Clinic, Rochester, MN, USA. [49]Department of Oncology and Metabolism, Sheffield Institute for Nucleic Acids (SInFoNiA), University of Sheffield, Sheffield, UK. [50]Academic Unit of Pathology, Department of Neuroscience, University of Sheffield, Sheffield, UK. [51]Department of Medical Epidemiology and Biostatistics, Karolinska Institutet, Stockholm, Sweden. [52]Department of Clinical Genetics, Fox Chase Cancer Center, Philadelphia, PA, USA. [53]Department of Pathology, Leiden University Medical Center, Leiden, The Netherlands. [54]Department of Human Genetics, Leiden University Medical Center, Leiden, The Netherlands. [55]Warwick Clinical Trials Unit, University of Warwick, Coventry, UK. [56]Department of Biomedical Sciences, Faculty of Science and Technology, University of Westminster, London, UK. [57]Department of Oncology, University of Cambridge, Cambridge, UK. [58]Cancer Sciences Academic Unit, Faculty of Medicine, University of Southampton, Southampton, UK. [59]Channing Division of Network Medicine, Department of Medicine, Brigham and Women's Hospital, Harvard Medical School, Boston, MA, USA. [60]Department of Epidemiology, Harvard T.H. Chan School of Public Health, Boston, MA, USA. [61]Department of Cancer Epidemiology, Clinical Sciences, Lund

University, Lund, Sweden. [62]Division of Evolution and Genomic Medicine, School of Biological Sciences, Faculty of Biology, Medicine and Health, Manchester Academic Health Science Centre, University of Manchester, Manchester, UK. [63]Genomic Medicine, St Mary's Hospital, Manchester Centre for Genomic Medicine, Manchester University Hospitals NHS Foundation Trust, Manchester Academic Health Science Centre, Manchester, UK. [64]NIHR Manchester Biomedical Research Centre, Manchester Academic Health Science Centre, Manchester University NHS Foundation Trust, Manchester, UK. [65]Division of Hematology and Oncology, Department of Medicine, David Geffen School of Medicine, University of California at Los Angeles, Los Angeles, CA, USA. [66]Usher Institute of Population Health Sciences and Informatics, The University of Edinburgh Medical School, Edinburgh, UK. [67]Cancer Research UK Edinburgh Centre, Edinburgh, UK. [68]Department of Breast Surgery, Herlev and Gentofte Hospital, Copenhagen University Hospital, Herlev, Denmark. [69]Genomic Medicine Group, Galician Foundation of Genomic Medicine, Instituto de Investigación Sanitaria de Santiago de Compostela (IDIS), Complejo Hospitalario Universitario de Santiago, SERGAS, Santiago de Compostela, Spain. [70]Moores Cancer Center, University of California San Diego, La Jolla, CA, USA. [71]Epidemiology Research Program, American Cancer Society, Atlanta, GA, USA. [72]Division of Genetics and Epidemiology, Institute of Cancer Research, London, UK. [73]Medical Oncology Department, Hospital Clínico San Carlos, Instituto de Investigación Sanitaria San Carlos (IdISSC), Centro Investigación Biomédica en Red de Cáncer (CIBERONC), Madrid, Spain. [74]Division of Genetics and Epidemiology, The Institute of Cancer Research, London, UK. [75]Cancer Epidemiology Division, Cancer Council Victoria, Melbourne, VIC, Australia. [76]Centre for Epidemiology and Biostatistics, Melbourne School of Population and Global Health, The University of Melbourne, Melbourne, VIC, Australia. [77]Department of Epidemiology and Preventive Medicine, Monash University, Melbourne, VIC, Australia. [78]Department of Dermatology, Huntsman Cancer Institute, University of Utah School of Medicine, Salt Lake City, UT, USA. [79]Department of Surgery, Oulu University Hospital, University of Oulu, Oulu, Finland. [80]Cancer & Environment Group, Center for Research in Epidemiology and Population Health (CESP), University Paris-Saclay, INSERM, University Paris-Sud, Villejuif, France. [81]Cardiovascular Epidemiology Unit, Department of Public Health and Primary Care, University of Cambridge, Cambridge, UK. [82]Department of Preventive Medicine, Keck School of Medicine, University of Southern California, Los Angeles, CA, USA. [83]Institute of Environmental Medicine, Karolinska Institutet, Stockholm, Sweden. [84]Molecular Genetics of Breast Cancer, German Cancer Research Center (DKFZ), Heidelberg, Germany. [85]Department of Medical Oncology, Family Cancer Clinic, Erasmus MC Cancer Institute, Rotterdam, The Netherlands. [86]Division of Cancer Sciences, University of Manchester, Manchester, UK. [87]Department of Surgery, National Taiwan University Hospital and National Taiwan University College of Medicine, Taipei, Taiwan. [88]Program in Genetic Epidemiology and Statistical Genetics, Harvard T.H. Chan School of Public Health, Boston, MA, USA. [89]Nuffield Department of Population Health, University of Oxford, Oxford, UK. [90]Department of Genetics and Pathology, Pomeranian Medical University, Szczecin, Poland. [91]Independent Laboratory of Molecular Biology and Genetic Diagnostics, Pomeranian Medical University, Szczecin, Poland. [92]Division of Oncology, Department of Medicine, Stanford Cancer Institute, Stanford University School of Medicine, Stanford, CA, USA. [93]Faculty of Medicine, University of Heidelberg, Heidelberg, Germany. [94]Radiation Epidemiology Branch, Division of Cancer Epidemiology and Genetics, National Cancer Institute, Bethesda, MD, USA. [95]Department of Surgical Oncology, Family Cancer Clinic, Erasmus MC Cancer Institute, Rotterdam, The Netherlands. [96]VIB, VIB Center for Cancer Biology, Leuven, Belgium. [97]Laboratory for Translational Genetics, Department of Human Genetics, University of Leuven, Leuven, Belgium. [98]Epidemiology Program, University of Hawaii Cancer Center, Honolulu, HI, USA. [99]Carmel Medical Center and Technion Faculty of Medicine, Clalit National Cancer Control Center, Haifa, Israel. [100]Department of Molecular Medicine and Surgery, Karolinska Institutet, Stockholm, Sweden. [101]Department of Clinical Genetics, Karolinska University Hospital, Stockholm, Sweden. [102]Institute of Clinical Medicine, Pathology and Forensic Medicine, University of Eastern Finland, Kuopio, Finland. [103]Department of Clinical Pathology, Imaging Center, Kuopio University Hospital, Kuopio, Finland. [104]Unit of Medical Genetics, Department of Medical Oncology and Hematology, Fondazione IRCCS Istituto Nazionale dei Tumori di Milano (INT), Milan, Italy. [105]Department of Oncology, Sšdersjukhuset, Stockholm, Sweden. [106]Department of Clinical Science and Education, Sšdersjukhuset, Karolinska Institutet, Stockholm, Sweden. [107]Department of Family Medicine and Public Health, University of California San Diego, La Jolla, CA, USA. [108]Department of Medical Oncology, University Hospital of Heraklion, Heraklion, Greece. [109]Department of Gynecology and Obstetrics, Ludwig Maximilian University of Munich, Munich, Germany. [110]Precision Medicine, School of Clinical Sciences at Monash Health, Monash University, Clayton, VIC, Australia. [111]Department of Laboratory Medicine and Pathobiology, University of Toronto, Toronto, ON, Canada. [112]Laboratory Medicine Program, University Health Network, Toronto, ON, Canada. [113]Department of Population Sciences, Beckman Research Institute of City of Hope, Duarte, CA, USA. [114]Department of Obstetrics and Gynecology, Helsinki University Hospital, University of Helsinki, Helsinki, Finland. [115]Department of Epidemiology, Lineberger Comprehensive Cancer Center, University of North Carolina at Chapel Hill, Chapel Hill, NC, USA. [116]Department of Health Sciences Research, Mayo Clinic, Rochester, MN, USA. [117]Centre for Cancer Research and Cell Biology, Queen's University Belfast, Belfast, Ireland, UK. [118]Genome Diagnostics Program, IFOM - the FIRC (Italian Foundation for Cancer Research) Institute of Molecular Oncology, Milan, Italy. [119]Research Oncology, Guy's Hospital, King's College London, London, UK. [120]Department of Oncology, Leuven Multidisciplinary Breast Center, Leuven Cancer Institute, University Hospitals Leuven, Leuven, Belgium. [121]Medical Oncology Department, Hospital Universitario Puerta de Hierro, Madrid, Spain. [122]Department of Oncology, University Hospital of Larissa, Larissa, Greece. [123]Center for Hereditary Breast and Ovarian Cancer, University Hospital of Cologne, Cologne, Germany. [124]Center for Molecular Medicine Cologne (CMMC), University of Cologne, Cologne, Germany. [125]Department of Gynaecology and Obstetrics, University Hospital Ulm, Ulm, Germany. [126]Genomics Center, Research Center, Centre Hospitalier Universitaire de Québec - Université Laval, Québec City, QC, Canada. [127]National Center for Tumor Diseases, University of Heidelberg, Heidelberg, Germany. [128]Department of Clinical Pathology, The University of Melbourne, Melbourne, VIC, Australia. [129]Division of Breast Cancer Research, The Institute of Cancer Research, London, UK. [130]Faculty of Medicine, University of Southampton, Southampton, UK. [131]Department of Genetics, Portuguese Oncology Institute, Porto, Portugal. [132]Biomedical Sciences Institute (ICBAS), University of Porto, Porto, Portugal. [133]Department of Epidemiology, Mailman School of Public Health, Columbia University, New York, NY, USA. [134]Peter MacCallum Cancer Center, Melbourne, VIC, Australia. [135]Sir Peter MacCallum Department of Oncology, The University of Melbourne, Melbourne, VIC, Australia. [136]Department of Surgery, Leiden University Medical Center, Leiden, The Netherlands. [137]Institute of Cancer and Genomic Sciences, University of Birmingham, Birmingham, UK. [138]Wellcome Trust Centre for Human Genetics and Oxford NIHR Biomedical Research Centre, University of Oxford, Oxford, UK. [139]Family Cancer Clinic, The Netherlands Cancer Institute - Antoni van Leeuwenhoek Hospital, Amsterdam, The Netherlands. [140]Biocenter Oulu, Cancer and Translational Medicine Research Unit, Laboratory of Cancer Genetics and Tumor Biology, University of Oulu, Oulu, Finland. [141]Laboratory of Cancer Genetics and Tumor Biology, Northern Finland Laboratory Centre Oulu, Oulu, Finland. [142]Department of Surgical Sciences, Uppsala University, Uppsala, Sweden. [143]Division of Molecular Carcinogenesis, The Netherlands Cancer Institute - Antoni van Leeuwenhoek Hospital, Amsterdam, The Netherlands. [144]Faculty of EEMCS, Delft University of Technology, Delft, The Netherlands. [145]The Donnelly Centre, University of Toronto, Toronto, ON, Canada. [146]Division of Psychosocial Research and Epidemiology, The Netherlands Cancer Institute - Antoni van Leeuwenhoek Hospital, Amsterdam, The Netherlands. [147]These authors jointly supervised this work: Sander Canisius, Marjanka K. Schmidt. *email: s.canisius@nki.nl; mk.schmidt@nki.nl

