## [Peer Review File · Nature Communications]

Reviewers' comments:

Reviewer #1 (Remarks to the Author):

Manuscript: Escala-Garcia et al.

What are the major claims of the paper?

In their manuscript, Escala-Garcia et al use a network approach to identify germline-driven gene modules associated with breast cancer survival. They claim to have identified a number of such modules: there are 3 high confidence GRPMs associated with ER-negative Breast cancer and 1 associated with ER-positive breast cancer. They use a validation data set to show that these modules are indeed transcriptionally deregulated in patients selected on the relevant genotypes. Are they novel and will they be of interest to others in the community and the wider field? In the past several GWAS have been carried out to identify genetic variants that are associated with disease progression. These studies have failed to find variants that are associated at genome-wide significance. Indeed, the authors repeat this analysis (Figure 2) and again fail to see an association. The authors then use the idea of combining variants that act in a common biological pathway, to achieve greater sensitivity. This has been applied in GWAS for disease and is applied here to the GWAS for disease progression.

The authors now find modules associated with disease. The identified modules fit with the general ideas the field has about processes that are thought to be associated with breast cancer progression but now give details on how these associations may be mediated.

Do you feel that the paper will influence thinking in the field?

Currently, personalised cancer treatments are primarily based on somatic mutations. However, the germline variants that contribute to progression may well be harnessed to further adapt treatments to the individual. The identification here of modules that are associated with disease progression may be a first step towards developing such personalised treatments or towards identifying new targets for treatment. The results are therefore of great interest to the wider breast cancer community.

Minor comments:

Line 432: "On closer inspection of the two modules' PHSs, we observed that the selected genetic variants for both modules only included variants in the same genomic region on chromosome 19p13.3 (Fig. 4a)."

This sentence is not very clear and could be rephrased.

Supplementary Figure 4: These plots are hard to understand on first reading. In the figure legend it would help to give some more detail, for example, to state how many gene scores were calculated and/or restate the window size on which each gene score was based.

Other comments:

I cannot comment on the details of the statistical analysis, but the authors of this paper have published widely on breast cancer GWAS and are experts in the statistical analysis of GWAS data.

Reviewer #2 (Remarks to the Author):

This is an intriguing study that identifies several network modules of common germline variants associated with poor survival in breast cancer in a large cohort of >84,000 breast cancer patients. A subset of the discovered associations are replicated in a separate cohort of ~12,000 patients. Identification of germline variants correlated with cancer prognosis is a difficult and unsolved problem, and thus the positive results reported in this manuscript are a welcome contribution to the field.

The bioinformatics analysis is straightforward and relies on two existing methodologies: the Pascal

method to compute gene-scores and the HotNet2 method to identify network modules. This use of existing methodology is a strong point of the manuscript, as these methods have been vetted externally. However, I have several reservations regarding some of statistical analyses and potential confounders.

1) The statistical analysis is almost exclusively described in terms of p-values without reporting effect sizes. Thus, a natural question arises regarding the magnitude of any of the reported effects. Are all of the reported effects extremely minor, but statistically significant because of the large sample sizes? The lack of effect sizes is particularly notable given the manuscript's focus on survival time where surprisingly not a single Kaplan-Meier is shown (with the exception of a schematic in Figure 1 giving an overview of the method).

2) I found the description of the gene expression analysis using TCGA data to be incoherent. How exactly do you "correlate[d] the PHS with the RNA expression data" as described in the Online Methods? Based on my understanding, the PHS gives a single score per patient with a small number of features, while RNA expression is available for all genes. I do not see how one computes a correlation between these two measurements.

3) The GNA11 and GNA15 result looks interesting, but it would be good to know more about the interactions in this module (see comment below regarding Figure 3). Do these genes interact in the PPI network, are there other published reports of interactions between these two genes, and/or do they share many interacting partners? The motivation for this question is that GNA11 and GNA15 are paralogs and the recording of protein-protein interactions for paralogous genes is notoriously unreliable with some PPI databases duplicating interactions without strong biological evidence. A similar problem could be arising with PER1 and PER3.

4) There is a subtle issue of ascertainment bias in network analysis that is not fully addressed in the current manuscript. Namely, cancer genes tend to be well studied and thus can have higher degree in a PPI network than less studied genes. Correlations between gene scores and gene degree are not preserved in the HotNet statistical test which can lead to inflated significance values. (See Hierarchical-HotNet, Reyna et al. Bioinformatics 2018). It would be good to check if there is an association between the Pascal derived gene scores and network degree, and if so to appropriate account for this confounder in the network analysis.

Additional minor comments:

Figure 3: This Figure had low resolution in PDF I reviewed. Text was fuzzy and some of gene names were unreadable. Please use high resolution (ideally vector graphics) for publication version. Also, while the superimposition of the results onto the Hallmarks is a nice visual, the authors should investigate potential copyright restrictions in the use of the Hallmarks graphics.

NCOMMS-19-02592

Point-by-point response to the reviewer's comments (responses in orange)

Reviewer #1 (Remarks to the Author):

Manuscript: Escala-Garcia et al.

What are the major claims of the paper?

In their manuscript, Escala-Garcia et al use a network approach to identify germline-driven gene modules associated with breast cancer survival. They claim to have identified a number of such modules: there are 3 high confidence GRPMs associated with ER-negative Breast cancer and 1 associated with ER-positive breast cancer. They use a validation data set to show that these modules are indeed transcriptionally deregulated in patients selected on the relevant genotypes.

Are they novel and will they be of interest to others in the community and the wider field?

In the past several GWAS have been carried out to identify genetic variants that are associated with disease progression. These studies have failed to find variants that are associated at genome-wide significance. Indeed, the authors repeat this analysis (Figure 2) and again fail to see an association. The authors then use the idea of combining variants that act in a common biological pathway, to achieve greater sensitivity. This has been applied in GWAS for disease and is applied here to the GWAS for disease progression.

The authors now find modules associated with disease. The identified modules fit with the general ideas the field has about processes that are thought to be associated with breast cancer progression but now give details on how these associations may be mediated.

Do you feel that the paper will influence thinking in the field?

Currently, personalised cancer treatments are primarily based on somatic mutations. However, the germline variants that contribute to progression may well be harnessed to further adapt treatments to the individual. The identification here of modules that are associated with disease progression may be a first step towards developing such personalised treatments or towards identifying new targets for treatment. The results are therefore of great interest to the wider breast cancer community.

Minor comments:

1) Line 432: “On closer inspection of the two modules’ PHSs, we observed that the selected genetic variants for both modules only included variants in the same genomic region on chromosome 19p13.3 (Fig. 4a).”

This sentence is not very clear and could be rephrased.

Response:

We thank the reviewer for this comment, and we apologize that the sentence was not clear. In order to give a clearer explanation, we have now rephrased the sentence to “On closer inspection of the genetic variants selected for the two modules’ PHSs, we observed that one genetic variant was shared by both modules. The other variants in the PHSs, two variants in total for the four-gene module and three variants for the module of 19 genes, were also located in the same genomic region on chromosome 19p13.3 (**Fig. 4a**). These variants are upstream of GNA11 in the former module and GNA15 in the latter.” (page 12 lines 453-457).

2) Supplementary Figure 4: These plots are hard to understand on first reading. In the figure legend it would help to give some more detail, for example, to state how many gene scores were calculated and/or restate the window size on which each gene score was based.

Response:

We appreciate this comment and acknowledge that the plots were not easy to understand at first. We have edited the plots to make them more comprehensible (see updated Supplementary Figure 4). We have removed the color gradient that was indicating density (or number of genes plotted at each location). We noticed that the coloring was not adding any relevant information and it was possibly making the figure harder to understand on first reading. We have also added additional information in the figure caption based on the reviewer's suggestions:

“Supplementary Figure 4: Scatter plots showing the $-\log_{10}$ P value of the $\sim 21,800$ gene scores computed within a 50-kb window-size around the gene region. Each dot represents a gene score. The correlations shown are Pearson correlations. (a) Estrogen Receptor (ER)-positive vs all breast cancers. (b) ER-negative vs all breast cancers. (c) ER-negative vs ER-positive breast cancers.”

Other comments:

I cannot comment on the details of the statistical analysis, but the authors of this paper have published widely on breast cancer GWAS and are experts in the statistical analysis of GWAS data.

Reviewer #2 (Remarks to the Author):

This is an intriguing study that identifies several network modules of common germline variants associated with poor survival in breast cancer in a large cohort of $>84,000$ breast cancer patients. A subset of the discovered associations are replicated in a separate cohort of $\sim 12,000$ patients. Identification of germline variants correlated with cancer prognosis is a difficult and unsolved problem, and thus the positive results reported in this manuscript are a welcome contribution to the field.

The bioinformatics analysis is straightforward and relies on two existing methodologies: the Pascal method to compute gene-scores and the HotNet2 method to identify network modules. This use of existing methodology is a strong point of the manuscript, as these methods have been vetted externally. However, I have several reservations regarding some of statistical analyses and potential confounders.

1) The statistical analysis is almost exclusively described in terms of p-values without reporting effect sizes. Thus, a natural question arises regarding the magnitude of any of the reported effects. Are all of the reported effects extremely minor, but statistically significant because of the large sample sizes? The lack of effect sizes is particularly notable given the manuscript's focus on survival time where surprisingly not a single Kaplan-Meier is shown (with the exception of a schematic in Figure 1 giving an overview of the method).

Response:

We thank the reviewer for this comment, and fully agree with it. We have included interval plots in the main results Figure 3 (see updated Fig. 3e) in order to report the effect sizes. The interval

plots illustrate the association of each GRPM's PHS with breast cancer specific-survival, both in the discovery and independent sets. Because the purpose of this manuscript is not measuring absolute survival, we only attach the Kaplan-Meier curves below for illustration, but prefer the interval plots for the paper. We would like to emphasize that the computation of the PHSs was used as an intermediate step or instrument to identify relevant biological networks and pathways, which is the main focus of the present manuscript.

Kaplan-Meier curves of the survival in the discovery set based on the PHSs for each of the five GRPMs. The two groups – PHS high and low, were split based on the median PHS value.

2) I found the description of the gene expression analysis using TCGA data to be incoherent. How exactly do you “correlate[d] the PHS with the RNA expression data” as described in the Online Methods? Based on my understanding, the PHS gives a single score per patient with a small number of features, while RNA expression is available for all genes. I do not see how one computes a correlation between these two measurements.

Response:

We apologize that the description of the method was not clear. We reformulated the method's description (page 23 lines 775-780) giving now a more detailed explanation of the correlation between PHS and expression data clearer: “For each GRPM, we computed the previously obtained PHS for the subset of TCGA patients with a tumor matching the subtype for which the GRPM was found. Next, we aimed to quantify the downstream transcriptional effect of the GRPM on the expression of every individual gene. To do so, we computed the Pearson correlation between the GRPM's PHS and the RNA expression of each gene. Finally, we

performed gene set enrichment analysis (GSEA)³² to test for enrichment of biological pathways among the highly correlating genes.”

3) The GNA11 and GNA15 result looks interesting, but it would be good to know more about the interactions in this module (see comment below regarding Figure 3). Do these genes interact in the PPI network, are there other published reports of interactions between these two genes, and/or do they share many interacting partners? The motivation for this question is that GNA11 and GNA15 are paralogs and the recording of protein-protein interactions for paralogous genes is notoriously unreliable with some PPI databases duplicating interactions without strong biological evidence. A similar problem could be arising with PER1 and PER3.

Response:

Indeed, GNA11 and GNA15 are paralogous. They do not interact in the PPI network used though, so there is no risk of a false positive interaction. In the case of GNA11 and GNA15, each gene encodes for a different alpha-subunit of the G-protein, which allows having several G-protein permutations. Related to possible interacting partners, there are indeed two shared interactors for GNA11 and GNA15, the genes CHRM2 and ADRB2. For all of those interactions, there appears to be experimental evidence, so they are not just inferred from the paralogous nature of GNA11 and GNA15.

On the other hand, PER1 and PER3 are both paralogous genes and interactors in the GRPM. Both genes have complementary functions regulating the circadian clock, and their interaction in the GRPM is supported by experimental evidence. As an example, it has recently been shown that PER3 is involved in the nuclear translocation of PER1 (Kazuhiro Yagita, Shun Yamaguchi, Filippo Tamanini, *et al.*).

4) There is a subtle issue of ascertainment bias in network analysis that is not fully addressed in the current manuscript. Namely, cancer genes tend to be well studied and thus can have higher degree in a PPI network than less studied genes. Correlations between gene scores and gene degree are not preserved in the HotNet statistical test which can lead to inflated significance values. (See Hierarchical-HotNet, Reyna et al. Bioinformatics 2018). It would be good to check if there is an association between the Pascal derived gene scores and network degree, and if so to appropriately account for this confounder in the network analysis.

Response:

We thank the reviewer for this valuable comment, because indeed we did not yet discuss ascertainment bias in the manuscript. At the reviewer’s request, we tested for potential correlations between the Pascal gene scores and the network degree. As can be seen in the scatterplots attached below (A), there was no correlation between the germline genetic gene scores and network degree in any of the analyzed subtypes. From this, we conclude that our analyses were not vulnerable to this potential source of ascertainment bias. For reference, we also tested whether gene scores obtained from pan-cancer somatic mutation data had a higher correlation with the network degree. As it can be seen in the scatterplots (B), the correlation is quite strong (Pearson $r^2 \sim 0.2$), suggesting that indeed, ascertainment bias relating to PPI node

degree is a valid concern with somatic mutation data, whereas in our germline analysis setting this does not seem an issue. Indeed, we acknowledge that using the newer method Hierarchical-HotNet would have been useful if there had been evidence of inflated values.

To clarify in the manuscript that there is no evidence for ascertainment bias in our dataset we have added a new paragraph in the discussion (pages 17-18 lines 620-631): “Using curated protein interaction networks such as iRefIndex in propagation analyses may cause a subtle type of ascertainment bias: more interactions tend to be known for better studied proteins, which proteins involved in tumor initiation and progression often are. As a result, gene scores may correlate positively with the number of interactions in the protein interaction network. This is the case, for example, when gene scores are based on somatic mutation frequencies in cancer. HotNet2 only controls for this partially, whereas a recent extension to the HotNet2 method provides a more rigorous solution⁷³. We tested whether our analysis was vulnerable to this ascertainment bias by calculating the correlation between the gene scores computed by Pascal and the number of interactions recorded by iRefIndex. For all, ER-positive, and ER-negative breast cancer, these correlations were close to zero (Pearson $r^2 = -0.012$, $r^2 = -0.006$, and $r^2 = 0.003$ respectively) showing no evidence of ascertainment bias due to proteins' numbers of recorded interactions.”

A) Gene scores based on genetic association with survival

B) Gene scores based on pan-cancer somatic mutations (mutation frequency or MutSig score)

Additional minor comments:

Figure 3: This Figure had low resolution in PDF I reviewed. Text was fuzzy and some of gene names were unreadable. Please use high resolution (ideally vector graphics) for publication version. Also, while the superimposition of the results onto the Hallmarks is a nice visual, the authors should investigate potential copyright restrictions in the use of the Hallmarks graphics.

Response:

We realize that the resolution in the PDF used in the first submission is low quality, apologies. Figures were made using Illustrator, therefore we have high resolution files that are now submitted with the reviewed files. Concerning any copyright restrictions, while the figure is indeed inspired by the Hallmarks of cancer graphics, we believe there is no copyright issue: the figures were completely made by us and the small symbol vectors included, we either created ourselves or took the clip-art available in Apple's Keynote. For the latter, we contacted Keynote support, which confirmed that we could freely use their symbols.

REVIEWERS' COMMENTS:

Reviewer #3 (Remarks to the Author):

This paper reports an interesting network based analysis of a breast cancer GWAS study that tries to identify germline variants associated with cancer progression (rather than cancer risk). This type of study is becoming increasingly important for designing precision or personalised oncology approaches. Our emerging experience with focussing on targeting the cancer driver genes suggests that - in many cases - it has limited success and also does not lend itself to personalization of therapy. We need to consider germline and network context. In this vein, the paper addresses some of the main limitations of GWAS studies, which are still suffering from the "is the glass half empty or half full?" syndrome when it comes to judging their success. Including network and pathway analysis possibly could break this impasse. This paper does both and I, therefore, regard it as important contribution to the field that earns publication in Nature Communications.

It seems that this paper has been reviewed before as a rebuttal is included. My main concerns are very similar to those of reviewer 2, and the authors have provided a rebuttal that largely addresses my concerns.

I have one additional comment, which briefly is mentioned in the paper but would deserve more attention. The network propagation uses generic protein interaction networks. It would be interesting to elaborate how tissue specific protein interaction network variations impinge on the results. I concede that this would require substantial more work, and maybe not all necessary data are available. However, if this cannot be reasonably done, it could be discussed in more depth.

NCOMMS-19-02592B

Point-by-point response to the reviewer's comments (responses in orange)

Reviewer #3 (Remarks to the Author):

This paper reports an interesting network based analysis of a breast cancer GWAS study that tries to identify germline variants associated with cancer progression (rather than cancer risk). This type of study is becoming increasingly important for designing precision or personalised oncology approaches. Our emerging experience with focussing on targeting the cancer driver genes suggests that - in many cases - it has limited success and also does not lend itself to personalization of therapy. We need to consider germline and network context. In this vein, the paper addresses some of the main limitations of GWAS studies, which are still suffering from the "is the glass half empty or half full?" syndrome when it comes to judging their success. Including network and pathway analysis possibly could break this impasse. This paper does both and I, therefore, regard it as important contribution to the field that earns publication in Nature Communications.

It seems that this paper has been reviewed before as a rebuttal is included. My main concerns are very similar to those of reviewer 2, and the authors have provided a rebuttal that largely addresses my concerns.

I have one additional comment, which briefly is mentioned in the paper but would deserve more attention. The network propagation uses generic protein interaction networks. It would be interesting to elaborate how tissue specific protein interaction network variations impinge on the results. I concede that this would require substantial more work, and maybe not all necessary data are available. However, if this cannot be reasonably done, it could be discussed in more depth.

Response:

We thank the reviewer for her/his positive feedback and appreciate the additional comment. We agree that elaborating further into how tissue specific interaction networks might affect the results is an interesting addition. Therefore, we have added a new paragraph in the discussion: "The modules that are identified also depend on the specificity of the PPI network to the disease-relevant tissue. Many proteins have tissue-specific expression patterns and functions; hence not all interactions in a generic PPI network are found in all tissues. The use of a tissue-specific PPI network may prevent discovery of false positive modules. One single most relevant tissue for our analysis is not easily identified though. Unlike the somatic mutations found in tumor cells, the germline variants we studied are present in every cell of the body. Their effect on survival may therefore be mediated by cell types or tissues other than the cancerous breast tissue. These include the various cell types present in the tumor microenvironment, or distant tissues that form the pre-metastatic niche. Furthermore, a PPI network specific for healthy breast tissue may not accurately describe the interactions active in transformed cancer cells. In our analysis, we used a generic PPI network. To prevent false positive modules, we complemented the network propagation with an extra filtering step in which we select high-confidence modules based on their association with survival." (pages 17-18 lines 617-629).